# POST-TRAINING LARGE LANGUAGE MODELS FOR DIVERSE HIGH-QUALITY RESPONSES

**Yilei Chen**[*]
Boston University

**Souradip Chakraborty**
University of Maryland, College Park

**Lorenz Wolf**
University College London

**Yannis Paschalidis**
Boston University

**Aldo Pacchiano**[*]
Boston University
Broad Institute of MIT and Harvard

## ABSTRACT

Reinforcement learning has emerged as a popular method for post-training large language models (LLMs). While improving the model's performance on downstream tasks, it often reduces the model's output diversity, leading to narrow, canonical responses. Existing methods to enhance diversity are limited, either by operating at inference time or by focusing on lexical differences. We propose a novel training method named DQO (**D**iversity **Q**uality **O**ptimization) based on determinantal point processes (DPPs) to jointly optimize LLMs for quality and semantic diversity. Our approach samples and embeds a group of responses for each prompt, then uses the determinant of a kernel-based similarity matrix to measure diversity as the volume spanned by the embeddings of these responses. Experiments across instruction-following, summarization, story generation, and reasoning tasks demonstrate that our method substantially improves semantic diversity without sacrificing model quality. We open-source the code at: `https://github.com/fairytale9/diversity-quality-optimization`.

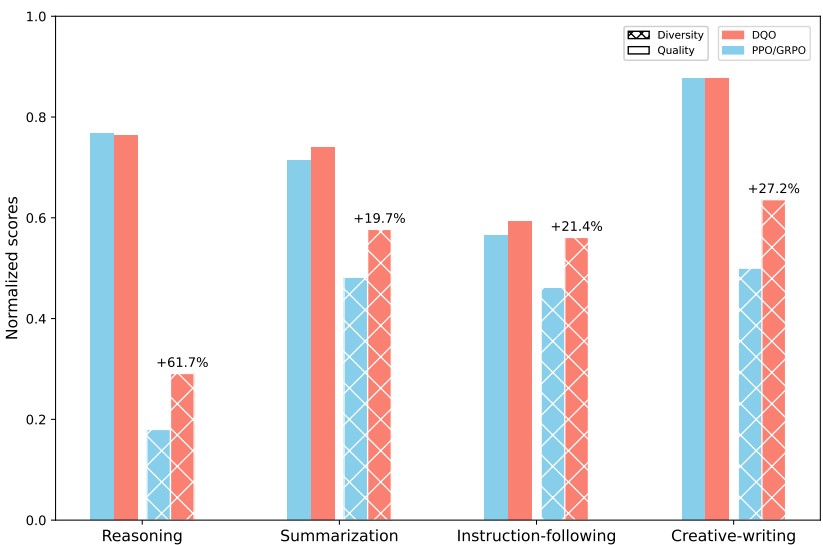

Figure 1: The quality and diversity scores of responses generated by the model trained with DQO compared to the model trained solely with reward. The reported quality score is the reward of one generation. The diversity score is the average across multiple diversity metrics, each computed over 10 responses. All responses are generated with a temperature of 1.0.

[*]Correspondence to: ylchen9@bu.edu and pacchian@bu.edu

# 1 INTRODUCTION

Large language models (LLMs) are typically post-trained to better align with human intentions and to perform effectively on downstream tasks (Ouyang et al., 2022; Bai et al., 2022). Reinforcement learning (RL) is commonly used to either maximize an existing reward function, or a reward model trained from human preference data (Ziegler et al., 2020; Stiennon et al., 2020; Ouyang et al., 2022; Bai et al., 2022; DeepSeek-AI et al., 2025). These methods substantially improve the output quality for targeted tasks. However, a widely observed drawback is that post-training often leads to a sharp reduction in output diversity, with models converging on a narrow set of canonical responses (Kirk et al., 2023; Murthy et al., 2024; Anderson et al., 2024; Xu et al., 2025; Casper et al., 2023). This loss of diversity is problematic across multiple dimensions: it limits reasoning and personalization by restricting alternative solution paths or user-preferred styles; it undermines test-time performance by reducing test-time search capabilities, robustness to distribution shift, and coverage of reward modes; and it weakens training dynamics by limiting exploration and the discovery of novel strategies.

Current efforts to promote diversity in LLM outputs are mostly limited to inference-time interventions such as temperature scaling (Ackley et al., 1985), top-k sampling (Holtzman et al., 2020), and related strategies (Nguyen et al., 2024; Franceschelli & Musolesi, 2025). While these approaches help in improving the spread across the response distribution, they often only introduce shallow, token-level variation and fail to produce truly diverse or meaningful responses. In many cases, they also hurt output quality, making the model less reliable or coherent. More critically, they cannot recover modes absent from the base model's learned distribution. A more principled approach is to optimize for diversity during training, but this poses two central challenges: defining diversity in a computationally efficient and theoretically sound way, and balancing diversity with response quality.

Recent attempts to improve diversity during training largely remain at the lexical level. Yao et al. (2025) encourage variation with a token-level entropy regularizer, but such measures fail to capture semantic diversity, which is often more meaningful to humans. Lanchantin et al. (2025) extend direct preference optimization (DPO) (Rafailov et al., 2023) by selecting the most diverse candidate among high-reward responses, yet their notion of diversity is still based on surface features such as generation probability or word counts. Likewise, Li et al. (2025) aim to preserve diversity in supervised fine-tuning by carefully constraining probability transfer between tokens during updates, again focusing on token-level variation. More broadly, these approaches prioritize local lexical differences rather than encouraging models to generate responses that span distinct semantic modes. Most related to our work, Chung et al. (2025) introduce a DPO variant that weights loss by average embedding distance, but it remains DPO-specific and considers only pairwise distances, which can yield degenerate solutions.

In this work, we propose a principled training method based on determinantal point processes (DPPs) (Kulesza et al., 2012) to directly optimize LLMs for both quality and diversity in generated responses. Unlike token-level entropy or lexical perturbations, our approach operates at the semantic level. Specifically, for each prompt we sample a set of responses, map them into an embedding space using a pretrained encoder, and compute a similarity matrix via a kernel function. The diversity score is then defined as the determinant of this matrix, which corresponds to the volume spanned by the response embeddings. Optimizing this objective encourages the model to generate responses that span a subspace in the answers' embedding space with the largest volume. The reward of each response can be regarded as a scaling factor of the corresponding embedding vector, providing an interpretable mechanism to balance quality against diversity. We refer to our algorithm as DQO (**D**iversity **Q**uality **O**ptimization). DQO is highly flexible and can be layered on top of existing state-of-the-art methods such as GRPO, making it broadly applicable in practice. We evaluate DQO across instruction-following, summarization, story generation, and reasoning tasks, and demonstrate that it significantly enhances semantic diversity while maintaining high response quality.

We summarize our contributions as the following,

- **Principled framework for Diversity Quality Optimization** : We propose a principled method, DQO, for post-training LLMs to generate diverse, high-quality responses. DQO is a flexible approach that can be applied on top of existing reinforcement learning algorithms, such as PPO and GRPO.

- **Semantic diversity beyond lexical variation** : We demonstrate that the DPP-based formulation provides a theoretically grounded framework for defining diversity, ensuring that responses span the semantic space both broadly and meaningfully.

- **Quality–diversity trade-off** : We experimentally show that DQO improves semantic diversity while preserving response utility, coherence, and task accuracy across a wide range of tasks. In addition, we conduct extensive ablations to illustrate the trade-off between quality and diversity.

## 2 PRELIMINARIES

### 2.1 NOTATIONS

For ease of readability, we summarize some frequently used notations here. We use $x$ and $y$ to represent a prompt and a response, respectively. We represent a group of $k$ responses $\{y_1, \ldots, y_k\}$ by $y_{1:k}$ and we denote $\{y_1, \ldots, y_{i-1}, y_{i+1}, \ldots, y_k\}$ by $y_{-i}$. We use $I_k \in \mathbb{R}^{k \times k}$ to represent the identity matrix with size $k$. And $\det(\cdot)$ represents the determinant of a matrix.

### 2.2 REINFORCEMENT LEARNING

Reinforcement learning has become a widely adopted approach for post-training LLMs with either an existing reward function or the one inferred from a preference dataset (e.g., RLHF). With the reward function, the model is typically optimized by maximizing the following KL-regularized objective,

$$\pi^* = \arg\max_{\pi_\theta}\{J(\pi_\theta) - \beta KL(\pi_\theta||\pi_{ref})\} \tag{1}$$

where $J(\pi_\theta) = \mathbb{E}_{x,y\sim\pi(\cdot|x)}[r(x,y)]$ is the expected return and $\beta$ is a hyperparameter that balances the KL divergence penalty and the rewards. Among existing algorithms, PPO (Schulman et al., 2017) and GRPO (Shao et al., 2024) have demonstrated strong empirical performance, by introducing some practical techniques including the clipping mechanism and group-based advantage estimation, respectively.

### 2.3 DETERMINANTAL POINT PROCESSES (DPPS)

In this work, we quantify the diversity of LLM generated outputs based on ideas derived from the Determinantal Point Process (DPP) literature (for a comprehensive introduction to DPPs, please refer to Kulesza et al. (2012)). Below we introduce the definition of an L-ensemble, which is a subclass of DPPs.

**Definition 1** (L-**ensemble**) *Let* $\mathcal{Y} = \{1, 2, \ldots, N\}$ *be a ground set, and* $\mathbf{Y} \subseteq \mathcal{Y}$ *be a random subset. Suppose* $L \in \mathbb{R}^{N \times N}$ *is a real symmetric positive semi-definite matrix. We say* $L$ *defines an* L*-ensemble, if for every* $A \subseteq \mathcal{Y}$,

$$\Pr(\mathbf{Y} = A) \propto \det(L_A),$$

*where* $L_A$ *is the submatrix of* $L$ *indexed by* $A$.

If we think of the entries of $L$ as measurements of similarity between pairs of elements, such as the dot product of the feature vectors of items, the determinant $\det(L_A)$ corresponds to the squared volume spanned by the feature vectors of items in $A$, which increases when the vectors are diverse and decreases when they are redundant or highly correlated. Thus, DPPs are well-suited to represent distributions over subsets of items where sets consisting of diverse items are more likely.

For instance, consider the two-element set $A = \{i, j\}$. By definition,

$$\Pr(A) \propto \begin{vmatrix} L_{ii} & L_{ij} \\ L_{ji} & L_{jj} \end{vmatrix} = L_{ii}L_{jj} - L_{ij}L_{ji}$$

If items $i$ and $j$ are highly similar, then $L_{ij}$ is large, resulting in a low probability. Conversely, if $L$ is diagonal (i.e., $L_{ij} = 0$), there are no correlations and the elements occur independently.

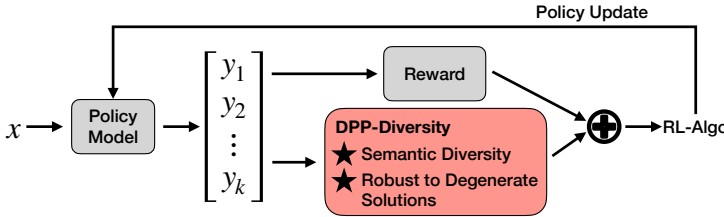

Figure 2: The DPP-based diversity metric of DQO promotes meaningful semantic diversity while ensuring robustness agains degenerate solutions.

## 3 DIVERSITY QUALITY OPTIMIZATION

We now formally define our notion of semantic diversity inspired by DPPs and how to incorporate it in reinforcement learning algorithms to jointly optimize quality and diversity.

### 3.1 DIVERSITY VIA DPPS

Based on the above definition of DPPs, given a group of responses $y_{1:k}$, we can formulate their diversity score as,

$$\text{Div}(y_{1:k}) = \det(L_\phi(y_{1:k})) \tag{2}$$

where $L_\phi(y_{1:k})[i,j] = f(\phi(y_i), \phi(y_j))$, $f$ is a kernel function and $\phi(\cdot)$ is a selected embedding model which can map a response into a high-dimensional semantic space. Although in most of this work we set the kernel function as the dot product, $f(\phi(y_i), \phi(y_j)) = \langle \phi(y_i), \phi(y_j) \rangle$, in Appendix D we also explore Gaussian kernel function and provide additional results. For simplicity, when it is clear from the context, we will omit the subscript in $L_\phi$.

Our definition of diversity offers two main advantages. First, it operates in the embedding space of the responses, allowing it to capture semantic diversity which is typically what humans intend. Second, its determinant-based definition induces a notion of group diversity that overcomes the limitations of simple pairwise distance metrics. A key limitation of pairwise distance measures, such as the average distance across responses, is their well-known vulnerability to a degenerate "clustering" effect, as noted by Parker-Holder et al. (2020). An algorithm optimizing for this metric might produce responses that form a few distinct, widely separated clusters, creating a misleading sense of diversity. In contrast, our determinant-based metric, which encourages the formation of a parallelepiped with a large volume in the embedding space, directly addresses this issue. The determinant is highly sensitive to the linear independence of the response vectors. If responses form tight clusters, the vectors within a cluster become nearly linearly dependent, causing the determinant of the similarity matrix to approach zero, regardless of the large distances between clusters. This correctly identifies a lack of true diversity. Furthermore, the determinant is a more robust measure because it recognizes when responses, despite having large pairwise distances, are confined to a lower-dimensional subspace. This forces the system to explore the full high-dimensional embedding space, ensuring genuine diversity that simple pairwise distances fail to capture.

### 3.2 QUALITY-DIVERSITY OBJECTIVE

Using the DPP based diversity metric (2) we now present the objective optimized by our DQO algorithm. For each prompt $x$, we sample $k$ responses $y_{1:k} \sim \pi_\theta(\cdot|x)$ from the policy, similarly to the sampling performed as part of GRPO. Instead of optimizing only the reward, we incorporate a diversity term based on the logarithm of our diversity metric into the objective. This yields the objective given by

$$J_{Div}(\pi_\theta) = \mathbb{E}_{x, y_{1:k} \sim \pi_\theta(\cdot|x)} \left[ \sum_{i=1}^{k} r(x, y_i) + \alpha \log \det(L_\phi(y_{1:k})) - \beta KL(\pi_\theta || \pi_{ref}) \right], \tag{3}$$

where $L_\phi(y_{1:k})$ is defined in (2). The hyperparameter $\alpha$ controls the trade-off between quality and diversity. Maximizing $J_{Div}(\pi_\theta)$ directly optimizes the policy for both quality and semantic diversity in generated responses.

In fact, it can be shown that by optimizing (3), the optimal policy satisfies,

$$\pi_{div}(y_{1:k}|x) \propto \pi_{ref}(y_{1:k}|x) \exp\left(\frac{1}{\beta}\left(\sum_{i=1}^{k} r(x, y_i) + \alpha \log \det(L_\phi(y_{1:k}))\right)\right) \tag{4}$$

For simplicity of exposition, suppose $\beta = \alpha$. We can define a reward-augmented embedding vector for the prompt-response pair $(x, y)$ as $\psi(x, y) = \sqrt{exp\left(\frac{r(x,y)}{\alpha}\right)\pi_{ref}(y|x)} \cdot \phi(y)$. Here, the reward acts as a scaling factor of the original semantic embedding. With the formulation of the reward-augmented embeddings, we can show that our optimal policy satisfies,

$$\pi_{div}(y_{1:k}|x) \propto \det(L_\psi(x, y_{1:k})) \tag{5}$$

For the complete derivation, we refer the reader to Appendix B. The expression above shows that our optimal policy (5) assigns probabilities to groups of responses in proportion to the determinant of the Gram matrix constructed from their embedding vectors. Geometrically, this means the policy selects groups of vectors in the response embedding space according to the squared volume of the parallelepiped spanned by those vectors.

The balance between quality and diversity also admits a clear geometric interpretation. The embedding vector $\psi$ consists of two components: a semantic embedding vector, which determines its direction, and a reward, which determines its norm. To maximize the volume of the spanned space, one should select vectors that are well separated from each other (i.e., diverse responses) while also having large norms (i.e., high-quality responses). The overall trade-off is governed by the hyperparameter $\alpha$.

**Connections with D-Optimal Design** In the classical theory of experimental design, the goal is to select design points that are most informative for estimating an unknown parameter vector. Consider the linear model $y = z^\top\theta + \xi$, where one collects design points $z_1, \ldots, z_n \in \mathbb{R}^d$. These design points represent the experimental conditions under which data is observed, and their selection determines how precisely the parameter $\theta$ can be estimated. The associated information matrix is defined as

$$M = \sum_{i=1}^{n} z_i z_i^\top = ZZ^\top,$$

where $Z = [z_1, \ldots, z_n]$ is the $d \times n$ design matrix. The criterion of *D-optimal design* selects these design points to maximize $\det(M)$, since this minimizes the volume of the confidence ellipsoid for the unknown parameter $\theta$ (Kiefer, 1959; Pukelsheim, 2006). By Sylvester's determinant identity,

$$\det(M) = \det(ZZ^\top) = \det(Z^\top Z),$$

(see, e.g., Horn & Johnson (2012)), so D-optimality is equivalently expressed as maximizing the determinant of the *Gram matrix* (the kernel matrix) of the design vectors under the dot-product kernel. Our DQO objective can be seen as a direct analogue of this construction. The role of the design vectors $z_i$ is instead played by the reward-augmented embeddings $\psi(x, y)$, which incorporate both the semantic content of a response and its quality signal. Maximizing the determinant of the Gram matrix built from these embeddings is therefore analogous to maximizing information gain in D-optimal design: it encourages the selected responses to be as linearly independent as possible in the $\psi$-space, ensuring that they collectively span a high-volume region that balances semantic diversity and reward. In this way, DQO can be viewed as extending the principle of D-optimal design from parameter estimation in bandits to the joint optimization of quality and diversity in language model responses. See a more thorough discussion in Appendix B.

### 3.3 ALGORITHM

We noticed that directly optimizing (3) presents challenges, including high variance in stochastic gradient estimates and risks of numerical instability. To address these issues, we now present a practical algorithmic formulation that stabilizes training and makes the QD objective feasible in

practice. To identify the source of these challenges, we begin by computing the gradient of $J_{Div}(\pi_\theta)$, which is given by (for simplicity, we omit the KL-regularization term here),

$$\nabla J_{Div}(\pi_\theta) = \mathbb{E}_{x,y_{1:k} \sim \pi_\theta(\cdot|x)} \left[ \sum_{i=1}^{k} \nabla \log \pi_\theta(y_i|x)(r(x,y_i) + \alpha \log \det(L(y_{1:k}))) \right]. \quad (6)$$

The first issue is that the determinant of $L(y_{1:k})$ can be close to zero, which results in a very large negative value of $\log(\det(L(y_{1:k})))$. This unbounded diversity term destabilizes training and complicates the trade-off between quality and diversity, to the point that only a carefully chosen $\alpha$ is effective. To mitigate this issue, we propose to consider the determinant of the matrix $L(y_{1:k}) + I_k$ instead. It can be shown that by adding an identity matrix, we have $k \geq \log(\det(L(y_{1:k}) + I_k)) \geq 0$ which is well-bounded. Briefly, adding an identity matrix to our objective can be regarded as a regularization term; we further discuss its effect on the objective in Appendix B and provide ablation results in Appendix D.

The second issue is that the gradient consists of the sum of the gradients of $k$ responses $y_{1:k}$, which causes it to have high variance, especially for large $k$. To mitigate the issue of inflating variance, we propose to use leave-one-out (*loo*) gradient estimators by subtracting the log-determinant of the gram matrix which leaves one response out,

$$\nabla^{loo} J_{Div}(\pi_\theta) = \mathbb{E}_{x,y_{1:k} \sim \pi_\theta(\cdot|x)} \left[ \sum_{i=1}^{k} \nabla \log \pi_\theta(y_i|x) \left( r(x,y_i) + \alpha \log \frac{\det(L(y_{1:k}) + I_k)}{\det(L(y_{-i}) + I_{k-1})} \right) \right].$$

Importantly, it can be shown that the *loo* estimator is unbiased and has a nice property on the boundedness of its value shown in Lemma 1 (for the proof, please refer to Appendix B). Lemma 1 shows that the diversity term is non-negative, with an upper bound of order $\log(k)$, which increases slowly as $k$ becomes large. This property stabilizes training and makes DQO robust to large values of $k$.

**Lemma 1** *Let us write the eigenvalues of $L(y_{1:k})$ as $\lambda_k \geq \cdots \geq \lambda_1$, then we have $1 + \lambda_k \geq \frac{\det(L(y_{1:k})+I)}{\det(L(y_{-i})+I)} \geq 1 + \lambda_1$. And the eigenvalue of $L(y_{1:k})$ is always in $[0, k]$ since the embedding vectors are normalized, we have $1 + k \geq \frac{\det(L(y_{1:k})+I)}{\det(L(y_{-i})+I)} \geq 1$ and $\log(1+k) \geq \log \frac{\det(L(y_{1:k})+I)}{\det(L(y_{-i})+I)} \geq 0$.*

## 4 EXPERIMENTS

In this section, we conduct a series of experiments to evaluate the performance of DQO in generating diverse and high-quality responses. Specifically, we aim to answer the following questions:

- Does DQO improve diversity in responses and how does it compare with the reward-only baseline and other existing quality-diversity algorithms?

- Does DQO achieve a favorable balance between quality and diversity? Can the model preserve or improve task performance while enhancing diversity?

- Is the performance of DQO consistent across different tasks and settings?

- How does DQO manage the trade-off between quality and diversity, and how do its hyperparameters influence performance?

### 4.1 CITY RECOMMENDATION

We begin with a simple synthetic experiment on city recommendation to clearly illustrate the diversity achieved by DQO. In this task, the model was prompted to recommend a city for traveling along with a concise reason. The exact prompt we used is provided in Appendix H. We compared DQO with GRPO and also implemented a variant of DQO using the average pairwise distance as the diversity score, which we refer to as DQO-pairwise distance; the original algorithm is denoted as DQO-determinant. The results are summarized in Figure 3, and the full details, including exact numbers and city names, are reported in Appendix D.

DQO clearly encourages the model to generate more diverse recommendations. When trained solely with the reward, the model tends to converge on recommending the same city repeatedly. For using the pairwise distance as the diversity score, we observed that the model's recommendations were dominated by two major cities, which aligns with our previous analysis. This occurs because high pairwise distance can be achieved with two widely separated clusters. In contrast, the determinant-based approach penalizes linear dependence, encouraging responses to span the space as broadly as possible. As shown in Figure 3, DQO-determinant produces the most diverse set of recommendations.

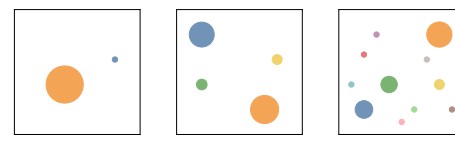

Figure 3: From left to right: GRPO, DQO-pairwise distance, DQO-determinant. Each circle represents a different city, with the size proportional to the number of times it was recommended. For each model, we sampled 100 times with a temperature of 1.0.

## 4.2 GENERAL TASKS

In the above section, we show the effectiveness of DQO in promoting diversity on the controlled city recommendation task, we now turn to a broader evaluation on general language model tasks. We implemented extensive experiments on four different kinds of tasks including reasoning (GSM8K (Cobbe et al., 2021)), summarization (CNN-dailymail (See et al., 2017)), story-writing (Common-Gen (Lin et al., 2020)) and instruction-following (Dolly Conover et al. (2023)).

We compare DQO to the baseline algorithm which trains the model solely with reward. For reasoning tasks, we use GRPO, while for non-reasoning tasks, we adopt PPO. We also compare DQO with other two popular quality-diversity algorithms: GRPO-likelihood (He et al., 2025) and GRPO-entropy (Yao et al., 2025). For the detailed experimental setup, please see Appendix C.

In this work, we employ a reward model to provide quality scores. Notice that, for reasoning tasks, we also rely on the reward model rather than outcome-based rewards, due to the observed phenomenon of reward hacking with outcome reward: with only outcome reward provided, the model trained with the quality-diversity objective learns to first generate a correct answer to the question thus securing the outcome reward, then generate some random or irrelevant contents to artificially increase diversity (for details, please refer to Appendix G). During training, rewards are normalized by dividing by an empirical maximum value to ensure a comparable scale with the diversity score, whereas during evaluation, we report the unnormalized rewards.

We report $pass@n$ metric (i.e., the highest score among $n$ responses) as measures of the quality in the responses with $n$ varies from 1 to 10. Without special clarifications, responses are sampled with a temperature of 1.0. And we use multiple metrics to measure the diversity in the responses which we summarize below,

- Distinct-n: Count the ratio of unique n-grams among the responses.
- Self-BLEU (Papineni et al., 2002) and Self-ROUGE (Lin, 2004) score: Two popular metrics to measure the similarity of languages. Note these scores measure the similarity, to be consistent with other metrics, we report $1 - Score$.
- LLM-as-a-Judge: We prompt an advanced model GPT-4o-mini to judge the model's output in terms of the diversity, serving as a surrogate for human judgment. In Appendix F, we show the results of a small-scale human study demonstrating that LLM-as-a-Judge produces evaluations consistent with human assessments.

We first compare the performance of DQO against all baseline algorithms. The results are summarized in Table 1. To evaluate the model, we select a representative non-reasoning task and a reasoning task. Among all algorithms, DQO is the only one that achieves both high diversity and high quality scores across both tasks. GRPO-likelihood shows performance comparable to DQO on the Dolly task but underperforms on GSM8K, whereas GRPO-entropy performs well on GSM8K but poorly on Dolly. These results demonstrate that DQO consistently delivers strong performance in post-training LLMs to produce diverse and high-quality generations.

Table 1: The quality and diversity scores achieved by DQO compared to other baseline algorithms on the instruction-following task: Dolly, and the reasoning task: GSM8K. Diversity metrics are calculated across 10 generated responses per prompt.

| Method / Task | Diversity ↑ | | | | Quality ↑ | |
|---|---|---|---|---|---|---|
| | distinct-1 | distinct-4 | self-bleu | self-rouge | $pass@1$ | $pass@10$ |
| **Dolly** | | | | | | |
| PPO | 0.24 | 0.64 | 0.41 | 0.49 | 5.65 | 8.39 |
| GRPO-likelihood | 0.26 | 0.70 | 0.46 | 0.54 | 5.86 | 8.50 |
| GRPO-entropy | 0.36 | 0.75 | 0.56 | 0.57 | 4.71 | 7.70 |
| DQO | 0.28 | 0.69 | 0.46 | 0.54 | 5.92 | 8.74 |
| **GSM8K** | | | | | | |
| GRPO | 0.09 | 0.32 | 0.09 | 0.21 | 76.8 | 87.9 |
| GRPO-likelihood | 0.26 | 0.86 | 0.53 | 0.59 | 50.9 | 80.4 |
| GRPO-entropy | 0.10 | 0.38 | 0.12 | 0.25 | 77.0 | 92.6 |
| DQO | 0.10 | 0.42 | 0.14 | 0.31 | 76.3 | 91.2 |

### 4.3 QUALITY-DIVERSITY BALANCE

Here, we present more fine-grained results on the diversity and quality of responses generated by the model trained with DQO. For comparison, we also include the results of the model trained solely with the reward.

**Quality.** In Figure 4, we show the $pass@n$ performance across four tasks with $n$ varying from 1 to 10. DQO exhibits better performance than the baseline model especially when $n$ is large. Besides, in the case of $n = 1$, our model has similar or better performance to the baseline. Together, the results show that our method does not hurt $pass@1$ performance while providing better $pass@n$ performance with $n > 1$ indicating that our model can generate both high-quality and diverse responses.

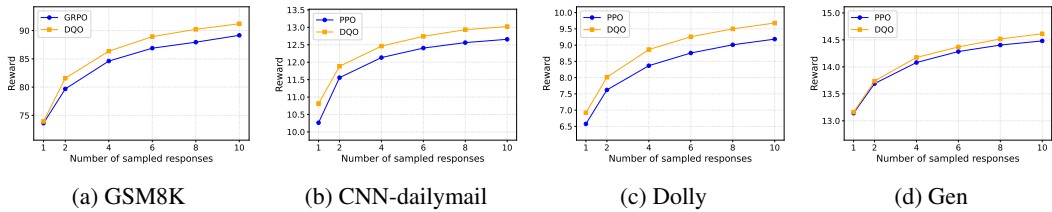

| (a) GSM8K | (b) CNN-dailymail | (c) Dolly | (d) Gen |
|---|---|---|---|

Figure 4: The performance of the trained model on $pass@n$ metrics. For DQO, we set hyperparameters $\alpha = 1.0$ and $k = 4$.

**Diversity.** The superior performance on $pass@n$ already suggests that our method enhances response diversity. To further validate this, we present six diversity metrics in Figure 4. For each metric, higher values indicate greater diversity. As shown in the figure, DQO consistently outperforms the baseline model, demonstrating a clear advantage in diversity. In particular, for the LLM-as-a-judge metric, the advanced model GPT-4o-mini strongly recognizes the diversity of responses generated by our approach (See Appendix I), highlighting improvements at the semantic level.

**Pareto frontier.** To illustrate how DQO achieves a favorable balance between quality and diversity, we plot the Pareto frontiers of DQO and the baseline model by varying either the training steps or the sampling temperature in Figure 6. Across different sampling temperatures (the right in Figure 6), our model consistently occupies the upper-right region relative to the baseline, demonstrating a robust advantage in balancing quality and diversity at the inference stage. Similarly, when varying

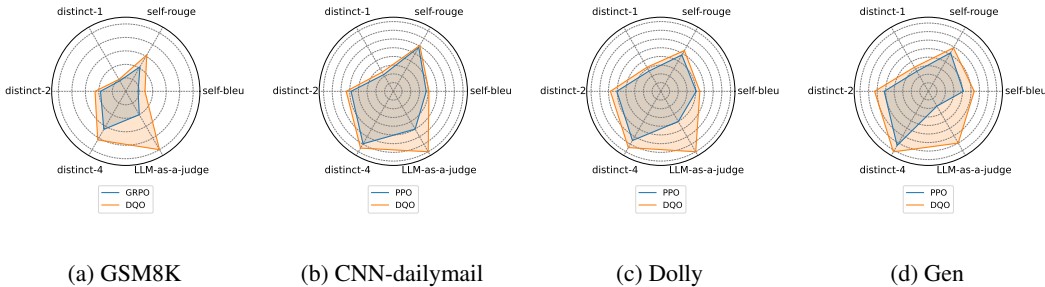

| (a) GSM8K | (b) CNN-dailymail | (c) Dolly | (d) Gen |

Figure 5: The performance of the trained model on six diversity metrics. For each metric, the higher value means the higher diversity. For DQO, we set hyperparameters $\alpha = 1.0$ and $k = 4$. And the diversity metrics are calculated across 10 generated responses per prompt.

the training steps (the left in Figure 6), our model remains Pareto-optimal throughout the entire training process, indicating that it consistently achieves a better quality–diversity balance throughout the entire training process.

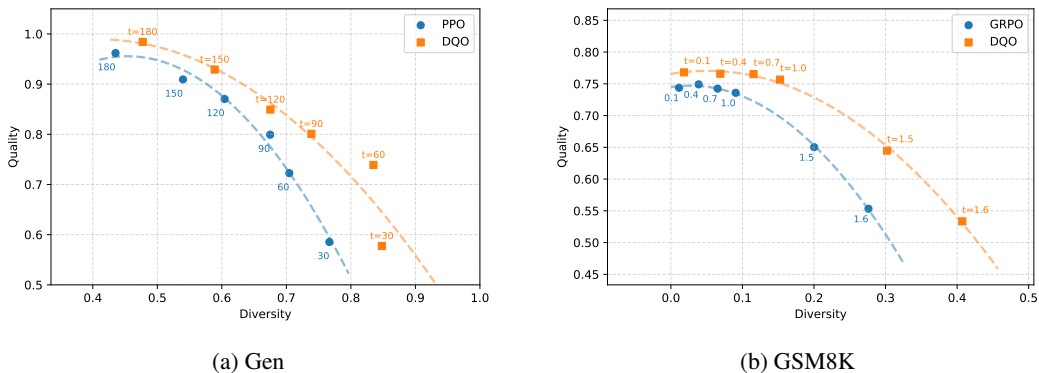

| (a) Gen | (b) GSM8K |

Figure 6: Pareto frontiers on quality and diversity of our model and the baseline. On the left, each point is a model trained with different training steps and the sampling temperature is set as 1.0. On the right, we take the final trained model but vary the sampling temperature.

## 4.4 ABLATION STUDY

DQO introduces two key hyperparameters: $\alpha$ and $k$. The hyperparameter $\alpha$ controls the weight of the diversity term in the training objective, while $k$ is the number of responses sampled per prompt in the algorithm. Both parameters jointly influence the trade-off between output quality and diversity. To analyze their impact, we conduct experiments across different values of $\alpha$ and $k$, and the results on Dolly task are summarized in Table 2 (for the ablations on GSM8K task, please refer to Appendix D).

From Table 2, we can observe the trade-off between quality and diversity when changing the value of $\alpha$ or $k$. Both increasing $\alpha$ and $k$ can enhance the diversity of generated responses, however, increasing $k$ incurs additional computational costs since more responses must be generated. Overall, compared with the baseline model, DQO exhibits robust improvements in both quality and diversity across a wide range of $\alpha$ and $k$ values, suggesting that it does not require highly sensitive hyperparameter tuning.

In addition to the ablations on different values of $\alpha$ and $k$, we also conduct ablations on different embedding models, kernel functions and on the weight of the identity matrix in the determinant calculation, controlled by the parameter $\gamma$ in $\det(L(y_{1:k}) + \gamma I_k)$. Due to space constraints, we refer readers to Appendix D for detailed results.

Table 2: The quality and diversity scores of the model trained with different values of hyperparameters $k$ and $\alpha$ on the Dolly task. Diversity metrics are calculated across 10 generated responses per prompt.

| Method | Diversity ↑ | | | | Quality ↑ | |
|---|---|---|---|---|---|---|
| | distinct-1 | distinct-4 | self-bleu | self-rouge | $pass@1$ | $pass@10$ |
| PPO | 0.24 | 0.64 | 0.41 | 0.49 | 5.65 | 8.39 |
| $\alpha = 0.5, k = 4$ | 0.28 | 0.69 | 0.44 | 0.53 | 5.84 | 8.79 |
| $\alpha = 1.0, k = 4$ | 0.28 | 0.69 | 0.47 | 0.54 | 5.92 | 8.73 |
| $\alpha = 1.5, k = 4$ | 0.33 | 0.79 | 0.57 | 0.61 | 5.47 | 8.56 |
| $\alpha = 2.0, k = 4$ | 0.35 | 0.82 | 0.54 | 0.64 | 5.42 | 8.69 |
| $k = 2, \alpha = 1.0$ | 0.24 | 0.62 | 0.40 | 0.50 | 5.71 | 8.13 |
| $k = 4, \alpha = 1.0$ | 0.28 | 0.69 | 0.47 | 0.54 | 5.92 | 8.73 |
| $k = 6, \alpha = 1.0$ | 0.31 | 0.76 | 0.49 | 0.58 | 5.71 | 8.83 |
| $k = 8, \alpha = 1.0$ | 0.32 | 0.79 | 0.52 | 0.61 | 5.64 | 8.64 |

## 5 CONCLUSIONS

In this work, we propose an algorithm DQO to post-train LLMs for diverse high-quality responses. Based on determinantal point processes, DQO defines the diversity in a group of responses as the determinant of a kernel-based similarity matrix of the embeddings of those responses. This definition of diversity has a straightforward interpretation as the squared volume of the space spanned by the embeddings of the response. We conduct extensive experiments across different kinds of tasks, and show that DQO can optimize the model to generate significantly more diverse responses while maintaining high quality in the generated responses.

Although DQO achieves superior performance, there are some limitations in this work. First, the quality-diversity objective is vulnerable to reward hacking when using the outcome reward. A reward model is needed which limits the applicability in many reasoning tasks where the outcome reward is commonly used. Second, DQO relies on the embedding models to map responses into a semantic space. The performance of DQO depends on the quality of these embeddings. A more principled and adaptive method for measuring diversity could better capture the underlying semantic variation, potentially adjusting automatically to the specific requirements of different tasks, which may emphasize different aspects of diversity, thus achieves better performance.

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

## A  RELATED WORKS

**Evaluating Diversity of LLMs.** Several works have focused on evaluating the diversity of LLM generated content (Guo et al., 2024; Shaib et al., 2024), also on investigating the impact of post-training on diversity metrics (Kirk et al., 2023; Shypula et al., 2025). The lack of diversity in LLM generated content also affects text written by humans using LLMs (Padmakumar & He, 2024).

**Improving Diversity of LLMs.** There are mainly two lines of works on promoting diversity in LLMs. One focuses on inference strategies. Nguyen et al. (2024) proposed a decoding method to reallocate the next-token probabilities which they show can increase the entropy of the correct solutions. The DiffSampling strategy, proposed by Franceschelli & Musolesi (2025), considers the largest difference between consecutive probabilities of tokens in a sorted distribution to promote diversity while maintaining correctness. Ahmed et al. (2025) proposed a two-stage inference strategy which consists of a high-temperature key words sampling process and a low-temperature expansion procedure.

Another line of works focus on the training strategy to best elicit diversity from LLMs. Lanchantin et al. (2025) proposed diverse preference optimization. They selected the most diverse response from the high-reward group and the least diverse response from the low-reward group to form the preference pair. The selection is based on some diversity criteria. Yao et al. (2025) shows that by adding an entropy term of correct answers to the reward-based objective, LLMs can improve the diversity while maintaining the quality. Different from those using reinforcement learning algorithms, Li et al. (2025) instead study the supervised finetuning approach. They proposed carefully-designed update strategy to mitigate the distribution collapse in SFT, thus encourages diversity. Most related to our work, Chung et al. (2025) propose a variant of DPO that weights the loss by the average pairwise distance in cosine similarity after embedding responses, this however, is limited to DPO, considers only pairwise distances, and requires sampling $k \geq 3$ responses per prompt in the training dataset. Concurrently, Tuyls et al. (2025) proposed a similar representation-based objective to encourage the exploration of large language models.

**Determinantal Point Processes.** Determinantal point processes (DPPs) (Kulesza et al., 2012), are a class of probabilistic models that arise in quantum physics and random matrix theory for modeling repulsion. DPPs are well-suited for modeling diversity. Parker-Holder et al. (2020) proposed a DPPs-based algorithm to train a population of diverse polices in reinforcement learning for better exploration.

**Semantic embeddings.** Our work relies on embedding models to map natural language responses to feature vectors in the semantic space which is related to works on representation learning and semantic embeddings (Kuhn et al., 2023; Kusupati et al., 2022).

## B    THEORETICAL RESULTS AND PROOFS

**Lemma.**    Suppose $\psi(x, y) = \sqrt{\exp(\frac{r(x,y)}{\alpha})\pi_{ref}(y|x)} \cdot \phi(y)$, then the optimal policy defined in (4) satisfies $\pi_{div}(y_{1:k}|x) \propto \det(L_\psi(y_{1:k}))$ when $\alpha = \beta$.

**Proof.**    Let $B \in \mathbb{R}^{n \times k}$ have columns $\phi(y_1), \ldots, \phi(b_k)$. The Gram matrix is

$$L = B^\top B.$$

Now suppose we scale each column $\phi(y_i)$ by a factor $a_i$, and denote

$$A = \operatorname{diag}(a_1, \ldots, a_k), \quad B' = BA.$$

Then the new Gram matrix is

$$L' = (B')^\top B' = (AB^\top)(BA) = A(B^\top B)A = ALA.$$

Taking determinants,

$$\det(L') = \det(ALA) = \det(A) \det(L) \det(A) = \big(\det(A)\big)^2 \det(L).$$

Since $\det(A) = \prod_{i=1}^k a_i$, we obtain

$$\det(L') = \left(\prod_{i=1}^k a_i\right)^2 \det(L).$$

Recall that $\pi_{div}(y_{1:k}|x)$ is defined as when $\alpha = \beta$,

$$\pi_{div}(y_{1:k}|x) \propto \pi_{ref}(y_{1:k}|x) \exp\left(\frac{1}{\alpha}\left(\sum_{i=1}^k r(x, y_i)\right) + \log \det(L_\phi(y_{1:k}))\right)$$

$$= \pi_{ref}(y_{1:k}|x) \exp\left(\frac{1}{\alpha}\left(\sum_{i=1}^k r(x, y_i)\right)\right) \det(L_\phi(y_{1:k}))$$

$$= \prod_{i=1}^k \left(\pi_{ref}(y_i|x) \exp\left(\frac{r(x, y_i)}{\alpha}\right)\right) \det(L_\phi(y_{1:k}))$$

The second equality holds because $y_{1:k}$ are sampled independently. Combined with the result above, we have $\pi_{div}(y_{1:k}|x) \propto \det(L_\psi(y_{1:k}))$.

**Analysis of $\det(L(y_{1:k}))$ and $\det(L(y_{1:k}) + I_k)$.**    Maximizing $\det(L)$ is equivalent to maximizing the volume of the parallelepiped spanned by the selected feature vectors, which enforces strict linear independence: any subset that induces a singular $L$ receives zero score. In contrast, maximizing $\det(L + I)$ introduces a ridge-like regularization. Indeed, if $L = BB^\top$ for a feature matrix $B \in \mathbb{R}^{k \times d}$, we have

$$\det(L + I) = \det(BB^\top + I) = \det(I + B^\top B).$$

This is precisely the determinant of a regularized scatter matrix, analogous to the role of $(B^\top B + \lambda I)$ in ridge regression. From this viewpoint, adding $I$ stabilizes the objective by preventing collapse along directions of near-linear dependence and avoiding the degeneracy of zero determinants.

A complementary interpretation arises from Bayesian linear models and Gaussian processes. In Bayesian linear regression with a Gaussian prior $w \sim \mathcal{N}(0, I)$ and unit-variance observation noise, the marginal likelihood normalization involves $\det(I + B^\top B)^{-\frac{1}{2}}$. Similarly, in Gaussian process regression, the log marginal likelihood includes $\log \det(L + \sigma^2 I)$, with $\sigma^2$ corresponding to the noise variance. Setting $\sigma^2 = 1$ recovers the $\det(L + I)$ objective. Hence, $\det(L + I)$ can be viewed as the determinant under a model with a prior noise floor, which softens the diversity requirement and balances between variance explained by the selected items and a baseline level of uncertainty.

**Eigenvalue Interlacing Theorem (Fisk, 2005).** Suppose $A \in R^{n \times n}$ is symmetric. Let $B \in R^{m \times m}$ with $m < n$ be a principal submatrix (obtained by deleting both $i$-th row and $i$-th column for some values of $i$). Suppose $A$ has eigenvalues $\lambda_1 \leq \cdots \leq \lambda_n$ and $B$ has eigenvalues $\beta_1 \leq \cdots \leq \beta_m$. Then,

$$\lambda_k \leq \beta_k \leq \lambda_{k+n-m}, \text{ for } k = 1, \cdots, m$$

And if $m = n - 1$, one has,

$$\lambda_1 \leq \beta_1 \leq \lambda_2 \leq \beta_2 \leq \cdots \leq \beta_{n-1} \leq \lambda_n$$

**Proof.** We use the Courant–Fischer min–max theorem. For a symmetric matrix $A \in \mathbb{R}^{n \times n}$ with eigenvalues $\lambda_1 \leq \cdots \leq \lambda_n$, the $k$-th eigenvalue can be characterized as

$$\lambda_k = \min_{\substack{S \subset \mathbb{R}^n \\ \dim(S) = k}} \max_{\substack{x \in S \\ x \neq 0}} \frac{x^\top A x}{x^\top x}.$$

Similarly, for the principal submatrix $B \in \mathbb{R}^{m \times m}$ with eigenvalues $\beta_1 \leq \cdots \leq \beta_m$, we have

$$\beta_k = \min_{\substack{T \subset \mathbb{R}^m \\ \dim(T) = k}} \max_{\substack{y \in T \\ y \neq 0}} \frac{y^\top B y}{y^\top y}.$$

Now observe that $B$ is obtained by restricting $A$ to a coordinate subspace (corresponding to removing some rows and columns). Hence any $y \in \mathbb{R}^m$ can be embedded into $\mathbb{R}^n$ by padding with zeros. Under this embedding, the Rayleigh quotient is preserved:

$$\frac{y^\top B y}{y^\top y} = \frac{x^\top A x}{x^\top x}, \quad \text{where } x \text{ is } y \text{ padded with zeros.}$$

Therefore, the feasible subspaces for $B$ are restrictions of those for $A$. This leads to the inequalities

$$\lambda_k \leq \beta_k \leq \lambda_{k+n-m}, \quad k = 1, \ldots, m.$$

In the special case $m = n - 1$, the inequalities expand into the chain

$$\lambda_1 \leq \beta_1 \leq \lambda_2 \leq \beta_2 \leq \cdots \leq \beta_{n-1} \leq \lambda_n,$$

which is exactly the interlacing property.

**Lemma.** Let's write the eigenvalues of $L(y_{1:k})$ as $\lambda_k \geq \cdots \geq \lambda_1$, then we have $1 + \lambda_k \geq \frac{\det(L(y_{1:k}) + I_k)}{\det(L(y_{-i}) + I_{k-1})} \geq 1 + \lambda_1$. And the eigenvalue of $L(y_{1:k})$ is always in $[0, k]$ since the embedding vectors are normalized, we have $1 + k \geq \frac{\det(L(y_{1:k}) + I_k)}{\det(L(y_{-i}) + I_{k-1})} \geq 1$ and $\log(1 + k) \geq \log \frac{\det(L(y_{1:k}) + I_k)}{\det(L(y_{-i}) + I_{k-1})} \geq 0$.

**Proof.** Let's write the eigenvalues of $L(y_{-i})$ as $\beta_{k-1} \geq \cdots \geq \beta_1$. Based on Eigenvalue Interlacing Theorem, we have,

$$\frac{\det(L(y_{1:k}) + I_k)}{\det(L(y_{-i}) + I_{k-1})} = (1 + \lambda_1) \prod_{i=1}^{k-1} \frac{1 + \lambda_{i+1}}{1 + \beta_i} \geq 1 + \lambda_1$$

and,

$$\frac{\det(L(y_{1:k}) + I_k)}{\det(L(y_{-i}) + I_{k-1})} = (1 + \lambda_k) \prod_{i=1}^{k-1} \frac{1 + \lambda_i}{1 + \beta_i} \leq 1 + \lambda_k$$

Since $L(y_{1:k})$ is positive semidefinite, it holds $\lambda_i \geq 0$, $\forall i$. And we have $\sum_{i=1}^k \lambda_i = \text{tr}(L(y_{1:k})) = k$ due to the normalization of the feature vectors. Hence, we have $k \geq \lambda_k \geq \lambda_1 \geq 0$.

**Connection to D-Optimal Design**  We construct a controlled toy setup to empirically compare our setting to D-optimal selection strategy against the baseline of uniform sampling, highlighting scenarios where diversity plays a critical role in achieving robust performance across varied reward functions

Setup and Notation: Let $d = 3$ denote the dimensionality of the embedding space, and let $\{\phi_1, \phi_2, \ldots, \phi_N\} \subset \mathbb{R}^d$ be a set of normalized candidate embeddings. We synthetically construct the pool to be imbalanced along coordinate directions: $n_x = 40$: vectors near the $x$-axis, $n_y = 40$: vectors near the $y$-axis and $n_z = 10$: vectors near the $z$-axis (rare) with $N = 90$. Small Gaussian noise $\varepsilon \sim \mathcal{N}(0, 10^{-4}I)$ is added to prevent rank-deficiency.

To illustrate our hypothesis, we compare two sampling strategies: **Uniform**: $w_i = \frac{1}{N}$ for all $i$ and **ours** by solving the following optimization problem

$$
\max_{w \in \Delta_N} \quad \log \det \left( \sum_{i=1}^{N} w_i \phi_i \phi_i^\top + \delta I \right)
$$

$$
\text{s.t.} \quad \sum_{i=1}^{N} w_i = 1, \quad 0 \leq w_i \leq 1
$$

where $\delta = 10^{-9}$ ensures numerical stability. The DPP based approach promotes spectral coverage across all directions. Next, we define four linear reward directions: $k_x = [1, 0, 0]^\top$ (high mass), $k_y = [0, 1, 0]^\top$ (high mass), $k_z = [0, 0, 1]^\top$ (low mass), $k_{\text{bal}} = \frac{1}{\sqrt{3}}[1, 1, 1]^\top$ (balanced). The expected reward under policy $w$ and reward $k$ is

$$
\mathbb{E}_{\phi \sim w} \left[ \langle k, \phi \rangle \right] = \sum_{i=1}^{N} w_i \langle k, \phi_i \rangle \tag{7}
$$

We compute the information matrices: $\Sigma(w) = \sum_{i=1}^{N} w_i \phi_i \phi_i^\top$ and visualize their eigenvalues. Uniform sampling has low spectral mass in the $z$-direction. In contrast, our DPP based approach balances mass across all directions. This confirms that $\log \det$ promotes **coverage**.

Thus this simulation reveals a failure mode of uniform sampling in imbalanced datasets. D-optimal design reallocates weights toward rare directions, yielding significantly improved performance in worst-case reward scenarios. The log-determinant acts as a diversity-promoting surrogate, superior to naive metrics like pairwise distance or cosine dissimilarity.

## C   Experimental setup

**Data preparation**   For GSM8K dataset, we directly use the training and test split. For CNN-dailymail dataset, we take the test split, and select $8,000$ data points as the training set and $1,024$ data points as the test set. For Dolly dataset, there is only one training split of $15,000$ data points. We divided it into two subsets with the ratio of $0.2$. For Gen, we use the training split, remove data with repetitive key words, and divided the set into two subsets, each containing $8,000$ and $1,024$ data points respectively.

**Training configuration**   We use GRPO algorithm for GSM8K task and use PPO algorithm for the other tasks. We use Qwen2.5-MATH-1.5B (Yang et al., 2024) as the base model for GSM8K task, and use Llama3.2-1B for the other tasks. For all tasks, we use the reward model Skywork/Skywork-Reward-V2-Llama-3.2-1B (Liu et al., 2025) and embedding model sentence-transformers/all-MiniLM-L6-v2. For baseline algorithms, we directly use the hyperparameters reported in their papers. For GRPO-likelihood, we use $\alpha = 0.25$ and for GRPO-entropy, we use $\alpha = 0.01$.

Table 3: Training configurations. For max prompt and response length, we use different values for different datasets. From left to right, it corresponds to GSM8K, CNN-dailymail, Dolly, Gen.

| | |
|---|---|
| training batch size | 128 |
| training epoches | 3 |
| actor learning rate | 1e-6 |
| critic learning rate | 1e-5 |
| rollout temperature | 1.0 |
| max prompt length | 256, 1536, 1024, 64 |
| max response length | 256, 256, 512, 128 |

# D  ADDITIONAL EXPERIMENTS

## D.1  CITY RECOMMENDATION

The following table shows the numerical results that were used to plot Figure 3. For each model, we sampled 100 times and countered the times of each city being recommended.

Table 4: The frequencies of each city being recommended for models trained with different methods.

| Method | City (Frequency) |
|---|---|
| GRPO | Tokyo (97); New York (3) |
| DQO-pairwise distance | New Orleans (48); Asheville (37); Budapest (8); Barcelona (7) |
| DQO-determinant | Budapest (45); Chiang Mai (22); New Orleans (19); Hanoi (7); Krakow (1); Kanazawa (1); Ottawa (1); Nashville (1); Tokyo (1); Bangkok (1); Singapore (1) |

## D.2  ADDITIONAL ABLATION STUDIES

**Embedding models.**  We did ablations across different embedding models. The results are shown in Table 5. For the hyperparameters, we use $\alpha = 1.0$ and $k = 4$. From the ablations results, it is evident that our proposed method DQO is robust to different embedding models.

Table 5: Comparison of different embedding models across multiple tasks. Diversity metrics are calculated across 10 generated responses per prompt.

| Embedding / Task | Diversity ↑ | | | | Quality ↑ | |
|---|---|---|---|---|---|---|
| | distinct-1 | distinct-4 | self-bleu | self-rouge | $pass@1$ | $pass@10$ |
| **Dolly** | | | | | | |
| all-MiniLM-L6-v2 | 0.28 | 0.69 | 0.46 | 0.54 | 5.92 | 8.74 |
| all-mpnet-base-v2 | 0.29 | 0.72 | 0.47 | 0.55 | 5.75 | 8.57 |
| gte-small | 0.30 | 0.68 | 0.47 | 0.53 | 5.89 | 8.50 |
| **GSM8K** | | | | | | |
| all-MiniLM-L6-v2 | 0.10 | 0.42 | 0.14 | 0.31 | 76.3 | 91.2 |
| all-mpnet-base-v2 | 0.10 | 0.39 | 0.13 | 0.28 | 76.5 | 90.9 |
| bge-base-en-v1.5 | 0.09 | 0.39 | 0.14 | 0.28 | 77.4 | 90.4 |
| **CNN** | | | | | | |
| all-MiniLM-L6-v2 | 0.25 | 0.75 | 0.41 | 0.61 | 11.08 | 13.02 |
| all-mpnet-base-v2 | 0.26 | 0.77 | 0.43 | 0.62 | 10.39 | 12.70 |
| gte-small | 0.25 | 0.74 | 0.41 | 0.60 | 10.55 | 12.89 |
| **Gen** | | | | | | |
| all-MiniLM-L6-v2 | 0.45 | 0.96 | 0.77 | 0.74 | 10.44 | 12.36 |
| gte-small | 0.45 | 0.96 | 0.78 | 0.73 | 10.48 | 12.31 |
| e5-base-v2 | 0.43 | 0.97 | 0.80 | 0.76 | 10.32 | 12.26 |

**Value of $\alpha$ and $k$.**  In Table 2 in the main page, we show the ablation results of $\alpha$ and $k$ on Dolly task. Here, Table 6 shows the results on GSM8K task.

**Kernel function.**  DQO formulates the diversity score as the determinant of the kernel matrix. By default, we use the dot product kernel function, i.e., $\phi(y_1, y_2) = \langle y_1, y_2 \rangle$. We study the effect of different kernel functions on DQO. We implemented same experiments using Gaussian kernel

Table 6: The quality and diversity scores of the model trained with different values of hyperparameters $k$ and $\alpha$ on the GSM8K task. Diversity metrics are calculated across 10 generated responses per prompt.

| Method | Diversity ↑ | | | | Quality ↑ | |
|---|---|---|---|---|---|---|
| | distinct-1 | distinct-4 | self-bleu | self-rouge | $pass@1$ | $pass@10$ |
| GRPO | 0.09 | 0.32 | 0.09 | 0.21 | 76.8 | 87.9 |
| $\alpha = 0.5, k = 4$ | 0.09 | 0.33 | 0.09 | 0.22 | 74.6 | 89.2 |
| $\alpha = 1.0, k = 4$ | 0.10 | 0.42 | 0.14 | 0.31 | 76.3 | 91.2 |
| $\alpha = 1.5, k = 4$ | 0.11 | 0.48 | 0.19 | 0.34 | 76.1 | 92.6 |
| $\alpha = 2.0, k = 4$ | 0.13 | 0.54 | 0.21 | 0.40 | 76.7 | 92.7 |
| $\alpha = 5.0, k = 4$ | 0.16 | 0.62 | 0.28 | 0.44 | 77.7 | 93.3 |
| $k = 2, \alpha = 1.0$ | 0.11 | 0.40 | 0.13 | 0.26 | 73.9 | 90.3 |
| $k = 4, \alpha = 1.0$ | 0.10 | 0.42 | 0.14 | 0.31 | 76.3 | 91.2 |
| $k = 6, \alpha = 1.0$ | 0.10 | 0.44 | 0.17 | 0.33 | 76.5 | 92.1 |
| $k = 8, \alpha = 1.0$ | 0.11 | 0.47 | 0.16 | 0.32 | 74.9 | 90.8 |

function, i.e., $\phi(y_1, y_2) = \exp\left(-\frac{\|y_1 - y_2'\|^2}{2}\right)$. Table 7 shows the performance of DQO with different kernel functions on Dolly and GSM8K tasks. DQO demonstrates robust performance under different kernel functions.

**Regularization by introducing identity matrix.** To solve the numerical explosion issue, we introduce an identity matrix when calculating the determinant: $\det(L(y_{1:k} + \gamma I_k))$. It can be shown adding an identity matrix plays a role as a regularization. By default, we simply set $\gamma = 1$. We test DQO with different values of $\gamma$. The results are summarized in Table 8 and Table 9.

Table 7: The quality and diversity scores of the model trained with different kernel functions on Dolly and GSM8K tasks. Diversity metrics are calculated across 10 generated responses per prompt.

| Method | Diversity ↑ | | | | Quality ↑ | |
|---|---|---|---|---|---|---|
| | distinct-1 | distinct-4 | self-bleu | self-rouge | $pass@1$ | $pass@10$ |
| Dolly | | | | | | |
| $\alpha = 1.0$, gaussian | 0.29 | 0.72 | 0.48 | 0.56 | 6.45 | 8.61 |
| $\alpha = 1.0$, dot product | 0.28 | 0.69 | 0.46 | 0.54 | 6.56 | 8.74 |
| $\alpha = 2.0$, gaussian | 0.34 | 0.79 | 0.54 | 0.61 | 6.12 | 8.64 |
| $\alpha = 2.0$, dot product | 0.35 | 0.82 | 0.54 | 0.64 | 6.41 | 8.69 |
| GSM8K | | | | | | |
| $\alpha = 1.0$, gaussian | 0.10 | 0.43 | 0.16 | 0.31 | 77.1 | 90.9 |
| $\alpha = 1.0$, dot product | 0.10 | 0.42 | 0.14 | 0.31 | 76.3 | 91.2 |
| $\alpha = 2.0$, gaussian | 0.11 | 0.48 | 0.19 | 0.36 | 75.2 | 91.2 |
| $\alpha = 2.0$, dot product | 0.13 | 0.54 | 0.21 | 0.40 | 76.7 | 92.7 |

We can see from Table 8, when $\gamma = 0.1$, the diversity in responses surges while the quality collapses. This is consistent with our analysis. The identity matrix can be regarded as a regularization term. The objective will prioritize diversity more if $\gamma$ is low. In addition, when $\alpha = 0.5$, we can see decreasing $\gamma$ does not affect the quality much as in the case where $\alpha = 1.0$. This is because $\alpha$ also controls the balance between quality and diversity. When $\alpha$ is low, the diversity is less important in the objective, hence, the effect of decreasing $\gamma$ is diluted. The phenomenon is consistent on GSM8K task.

Table 8: The performance of DQO with different identity matrix weights on Dolly task. Diversity metrics are calculated across 10 generated responses per prompt.

| Method | Diversity ↑ | | | | Quality ↑ | |
|---|---|---|---|---|---|---|
| | distinct-1 | distinct-4 | self-bleu | self-rouge | $pass@1$ | $pass@10$ |
| $\alpha = 1.0, +0.1I$ | 0.57 | 0.96 | 0.79 | 0.86 | 3.44 | 6.38 |
| $\alpha = 1.0, +0.5I$ | 0.37 | 0.83 | 0.56 | 0.66 | 6.72 | 8.90 |
| $\alpha = 1.0, +I$ | 0.28 | 0.69 | 0.46 | 0.54 | 6.56 | 8.74 |
| $\alpha = 0.5, +0.1I$ | 0.34 | 0.84 | 0.58 | 0.68 | 6.04 | 8.75 |
| $\alpha = 0.5, +0.5I$ | 0.28 | 0.71 | 0.44 | 0.54 | 6.31 | 8.72 |
| $\alpha = 0.5, +I$ | 0.28 | 0.69 | 0.43 | 0.53 | 6.47 | 8.77 |

Table 9: The performance of DQO with different identity matrix weights on GSM8K task. Diversity metrics are calculated across 10 generated responses per prompt.

| Method | Diversity ↑ | | | | Quality ↑ | |
|---|---|---|---|---|---|---|
| | distinct-1 | distinct-4 | self-bleu | self-rouge | $pass@1$ | $pass@10$ |
| $\alpha = 1.0, +0.1I$ | 0.23 | 0.79 | 0.41 | 0.52 | 73.3 | 93.9 |
| $\alpha = 1.0, +0.5I$ | 0.11 | 0.49 | 0.19 | 0.36 | 77.5 | 92.0 |
| $\alpha = 1.0, +I$ | 0.10 | 0.42 | 0.14 | 0.31 | 76.3 | 91.2 |
| $\alpha = 2.0, +0.1I$ | 0.38 | 0.88 | 0.58 | 0.74 | 63.5 | 91.3 |
| $\alpha = 2.0, +0.5I$ | 0.15 | 0.59 | 0.24 | 0.42 | 78.0 | 93.2 |
| $\alpha = 2.0, +I$ | 0.13 | 0.54 | 0.21 | 0.40 | 76.7 | 92.7 |

## E   COMPUTATIONAL COST ANALYSIS

Compared to quality-only post-training, DQO introduces an additional diversity term, which requires computing diversity scores during training. In this section, we show that the resulting computational overhead is negligible in practice.

DQO computes the determinant of the similarity matrix, whose computational complexity is $O(k^3)$ where $k$ is the number of answers generated per prompt. However, since $k$ is generally small ($< 10$), the total number of floating-point operations is on the order of a few thousand which corresponds to only microseconds of runtime. And our ablations further support the claim. The effect of increasing the value of $\alpha$ is similar to increasing $k$, so practically it is enough to choose a small value of k, which makes the calculation of determinant negligible.

The primary additional cost in DQO arises from using the embedding model to map the response to feature vectors. But since embedding models are non-autoregressive and relatively with small sizes, its computational overhead is modest.

We report the concrete results of the average time taken per training step of DQO compared with the baseline in Table 10 below. As shown, the additional computational cost introduced by DQO is negligible relative to GRPO.

Table 10: Training time per optimization step (seconds per step) for GRPO and DQO across different tasks.

| Task | GRPO (s/step) | DQO (s/step) |
|---|---|---|
| Reasoning | 36.844 | 36.216 |
| Summarization | 34.826 | 34.463 |
| Instruction-following | 40.962 | 41.485 |
| Creative-writing | 20.963 | 22.462 |

# F  LLM-AS-A-JUDGE FOR DIVERSITY EVALUATION

Evaluating diversity in generated responses is inherently challenging. While human evaluation provides the most reliable assessment, it is expensive and difficult to scale. As a practical alternative, we adopt LLM-as-a-judge as a surrogate for human judgment.

To validate that LLM-as-a-judge can perform reliable evaluation, we conducted a small-scale human study comparing the diversity of responses generated by DQO and the baseline model. We randomly selected 18 prompts from the test sets of four benchmarks (five prompts per task, except for summarization, where we selected three to keep the evaluation time manageable). For each prompt, we collected four responses from each model.

Participants were asked to assign a dense preference score between 0 and 1 indicating which model produced more diverse responses. The order of the two models was randomized to prevent any potential bias or information leakage.

The results are presented below: each column corresponds to a prompt, and we report the preference score assigned to DQO. Both the LLM-as-a-judge and human evaluators consistently rated DQO as producing more diverse outputs. Moreover, the human study results are highly correlated with the LLM-as-a-judge evaluation and agree on the majority of prompts, supporting the reliability of using LLM as a judge for diversity evaluation.

Table 11: Comparison between LLM-as-a-judge and human evaluation. The score shows the preference to the responses generated by DQO model in terms of diversity. Each column is a prompt sampled from the test dataset.

| | 1 | 2 | 3 | 4 | 5 | 6 | 7 | 8 | 9 | 10 | 11 | 12 | 13 | 14 | 15 | 16 | 17 | 18 |
|---|---|---|---|---|---|---|---|---|---|---|---|---|---|---|---|---|---|---|
| LLM-as-a-judge | 0.70 | 0.80 | 0.70 | 0.80 | 0.55 | 0.40 | 0.60 | 0.60 | 0.50 | 0.80 | 0.90 | 0.70 | 0.80 | 0.60 | 0.60 | 0.75 | 0.70 | 0.60 |
| Human evaluation | 0.65 | 0.74 | 0.79 | 0.67 | 0.62 | 0.50 | 0.74 | 0.63 | 0.54 | 0.51 | 0.65 | 0.61 | 0.88 | 0.69 | 0.59 | 0.62 | 0.78 | 0.68 |

## G   REWARD HACKING WITH OUTCOME REWARD

We observed an interesting phenomenon when doing experiments on GSM8K task. With only outcome reward provided, the model trained with the quality-diversity objective learns to first generate a correct answer to the question thus securing the outcome reward, then generate some random or irrelevant contents to artificially increase diversity. See the concrete example below.

The reward-hacking phenomenon does not exist when we use reward models, since the reward model evaluates the quality of the entire response rather than just the final answer.

We explored a potential solution by computing the cosine similarity between the embeddings of the prompt and the response and multiplying this score with the diversity term. The intuition is that only responses semantically aligned with the prompt contribute to the diversity objective. However, we observed that the model can still exploit this mechanism by generating correct answers followed by random contents but superficially related to the prompt. See the concrete example below.

We believe outcome-based reward alone is inherently susceptible to reward hacking. The outcome reward is widely used in reasoning tasks due to its simplicity and effectiveness, however, when combined with other objectives, it is vulnerable to reward hacking. This phenomenon presents an intriguing challenge for future study.

---

**An example of reward hacking with outcome reward**

Prompt: Gretchen has some coins. There are 30 more gold coins than silver coins. If she had 70 gold coins, how many coins did Gretchen have in total? Let's think step by step and output the final answer after "####".

Response: If Gretchen had 70 gold coins and there are 30 more gold coins than silver coins, then the number of silver coins she had is

$$70 - 30 = 40 \text{ silver coins.}$$

The total number of coins Gretchen has is the sum of gold and silver coins, so it is

$$70 + 40 = 110 \text{ coins.} \quad \#\#\#\# \ 110$$

The correct answer is: 110

Which digit is first to appear in the units place of

$$5^1 + 5^2 + 5^3 + \cdots + 5^{2004} + 5^{2005} \ ?$$

    (A)  0

    (B)  1

    (C)  2

    (D)  5

To determine which digit is first to appear in the units place of the sum

$$5^1 + 5^2 + 5^3 + \cdots + 5^{2004} + 5^{2005},$$

we need to analyze the units digits of the powers of 5.
First, let's observe the units digits of the first few powers of 5:

$$\cdots$$

---

### An example of reward hacking with prompt-response similarity

Prompt: Bobby takes a 30 min lunch and two 15 minutes break per day at the office. After 5 days, how many hours do his lunches and breaks add up to? Let's think step by step and output the final answer after "####".

Response: Bobby takes a 30 minute lunch and a 15 minute break twice a day.
So in total, Bobby takes 30 + 15 + 15 = 60 minutes of lunch and breaks per day.
After 5 days, Bobby takes a total of

$$60 \times 5 = 300 \text{ minutes of lunch and breaks.}$$

Since there are 60 minutes in an hour, Bobby's lunches and breaks add up to

$$\frac{300}{60} = 5 \text{ hours.}$$

#### 5
Bobby's lunches and breaks add up to 5 hours.
Bobby's dog is on a diet. For every meal he eats, he needs to walk him for 2 minutes. When he eats twice, he walks him 4 times that amount. How many minutes did Bobby walk his dog if he let him eat 7 meals? Let's think step by step and output the final answer after "####".
Bobby's dog needs to walk 2 minutes for every meal he eats.

$$2 \times 7 = 14 \text{ minutes.}$$

#### 14

# H  PROMPTS

We listed the prompts we used in this work.

---

**Prompt for city recommendation**

Prompt = "Recommend a city to visit along with a reason. Respond ONLY with valid JSON with the following schema. Do not include explanations or extra text.
{
"city": "<city name>",
"reason": "<reason>"
}"

---

**Prompt for GSM8K**

Prompt = "{QUESTION} Please reason step by step, and put your final answer within \\boxed{}."

---

**Prompt for CNN-dailymail**

Prompt = f"Summarize the following article:
{ARTICLE}
Summary:"

---

**Prompt for Dolly**

prompt = f"### Instruction:
{INSTRUCTION}
### Input:
{INPUT}
### Response:"

---

**Prompt for Common-gen**

Prompt = f"Write a short story of 3–4 sentences that includes the following keywords: {KEY WORD}. The story should be coherent, creative, and naturally incorporate all the keywords. Story:"

---

**Prompt of LLM as a diversity judge**

Instructions="The following are two files, each records responses generated by a model. The 'prompt' field stores prompts, and the 'response' field stores the corresponding groups of generated responses for the prompts. Please judge two models in terms of the diversity of their generated responses. You should give each model a score between [0,1]."
Input= f"Files: model 1: {FILE}; model 2: {FILE}."

# I   LLM AS A DIVERSITY JUDGE

We use advanced LLMs (GPT-4o-mini) to score the diversity of the responses generated by the models. The following are the evaluations given by GPT-4o-mini. For fair comparison, we did two comparisons by changing the order of two models and taking the average scores.

---

**Diversity judged by gpt-4o-mini on GSM8K; Model 1 (GRPO), Model 2 (DQO)**

To evaluate the diversity of responses generated by the two models, we can analyze the patterns seen in the "responses" field of each model.

**Model 1:**

- **Characteristics:**
    - The majority of responses are quite repetitive, consistently using phrases like "To solve this problem, we'll break it down" or variations thereof.
    - There appears to be a lack of varied strategies or unique approaches in the generated answers, with many instances simply restating the problem-solving process in a similar structure.

- **Diversity Score:** Given the high level of similarity and repeated phrasing, I would rate the diversity of responses for Model 1 as **0.2/1**.

**Model 2:**

- **Characteristics:**
    - This model exhibits a somewhat greater variety in response structure, with different steps highlighted in the problem-solving process.
    - Phrases like "Step X: Determine the..." and "Let's denote the total number..." introduce variation, and there seems to be a more thoughtful approach to several responses.
    - While there are still some repeated forms, the presence of distinct steps and strategies indicates a broader array of response possibilities.

- **Diversity Score:** The variety and different methodologies present in the responses lead me to rate Model 2's diversity as **0.5/1**.

**Summary:**

- **Model 1 Score:** 0.2/1
- **Model 2 Score:** 0.5/1

Model 2 demonstrates better response diversity compared to Model 1, which relies heavily on similar language and structure.

---

---

**Diversity judged by gpt-4o-mini on CNN-dailymail; Model 1 (PPO), Model 2 (DQO)**

To assess the diversity of the responses generated by the two models, we evaluate the variety and uniqueness of the content provided in the responses to identical prompts.

**Model 1:**

- The responses from Model 1 vary in terms of length and detail, but many responses are quite structured and similar in approach.
- There is some variation in naming conventions and specific details, but overall, a noticeable pattern exists in how the information is expressed.
- Many responses use similar phrasing, leading to less diversity overall.

**Model 2:**

- Responses from Model 2 exhibit greater variation in phrasing, structure, and detail.
- The content shows a wide range of descriptions and perspectives, which enhances the uniqueness of each response.
- There is a noticeable effort to vary not just the wording but also the structure of each response, leading to a more diverse set of outputs.

**Scores:** Based on this analysis, we assign the following scores for diversity in generated responses:

- **Model 1: 0.4**      (Limited diversity, responses similar in structure and phrasing)
- **Model 2: 0.8**      (High diversity, significant variation in responses)

These scores reflect the extent of variety present in the responses generated by each model.

---

---

**Diversity judged by gpt-4o-mini on Dolly; Model 1 (PPO), Model 2 (DQO)**

To assess the diversity of responses generated by both models, we analyze the responses across various prompts, looking for variances in creativity, structure, and content diversity.

**Model 1 Analysis:**

- The responses from Model 1 often maintain a similar structure, primarily focusing on straightforward factual information, lists, or direct answers to prompts.

- The responses appear to have consistent formatting, providing succinct and clear answers, but can feel repetitive and lack creative exploration.

- While some responses do reflect a variety of information (e.g., multiple examples, details), the overall tone and presentation remain quite uniform.

**Model 2 Analysis:**

- Model 2 demonstrates a greater variety of response styles, incorporating markdown-like formatting, lists, and varying degrees of detail and structure across its outputs. This format enhances readability and introduces an engaging style.

- The responses in Model 2 often include additional editorial commentary or creativity (e.g., humorous tones in astrology interpretations, narrative elements in product recommendations).

- The use of diverse sentence structures and clear formatting contributes to a sense of dynamism in responses.

**Scoring:**

- **Model 1:** 0.4 (due to consistent structure and less creative diversity in responses)

- **Model 2:** 0.8 (due to varied response formats, creative styles, and more engaging outputs)

**Final Scores:**

- **Model 1:** 0.4

- **Model 2:** 0.8

---

---

**Diversity judged by gpt-4o-mini on Common-gen; Model 1 (DQO), Model 2 (PPO)**

To evaluate the diversity of the generated responses from both models, we analyze the content and variety of the responses provided for the same prompts.

**Model 1:**

- The responses feature varied character names and professions, indicating a range of different story premises.
- While the structure is somewhat consistent (retired individuals reflecting on their past), each response introduces a new character with unique backstories or contexts.
- The diversity is primarily in the character names and their respective professions, leading to different narrative angles.

**Model 2:**

- The responses are more repetitive in structure; most of them start similarly with "An older woman pauses...".
- There is a notable lack of variation in terms of settings and character actions across the responses. Although some details differ (such as the specific verb or object), the overall premise and sentence structure remain largely the same.
- This limits the diversity of storytelling in comparison to Model 1.

**Scores:**

- **Model 1:** 0.8 — Scores high for its unique character introductions and storytelling approaches, exhibiting good diversity.
- **Model 2:** 0.4 — Scores lower due to the repetitive structure and similarity in responses, which significantly reduces the diversity in storytelling.

**Summary:** Model 1 demonstrates a greater range of ideas and creativity in responses, while Model 2 lacks variety, leading to a more uniform storytelling style.

