# OpenReview forum: "Post-training Large Language Models for Diverse High-Quality Responses"
_ICLR.cc/2026/Conference — ICLR 2026 Poster_

### Official Review · Reviewer_RQPB · 2025-10-24

**Soundness:** 3
**Presentation:** 2
**Contribution:** 3
**Rating:** 6
**Confidence:** 3

**Summary:**

This paper introduces DQO, a principled framework for post-training LLMs to optimize both diversity and quality of responses. Key contributions include: a flexible method compatible with existing RL algorithms like PPO and GRPO; a DPP-based formulation that ensures semantic diversity beyond lexical variation; and experimental demonstration of an effective quality-diversity trade-off across multiple tasks while maintaining utility and coherence.

**Strengths:**

1. The paper proposes a post-training reinforcement learning method from the perspective of diversity in content generated by LLMs, which simultaneously ensures generation quality and diverse semantic capabilities. Its research perspective and solution approach are innovative.

2. The paper presents a clear dual objective function for optimizing LLMs to ensure both quality and diversity, while also addressing two challenging issues in optimizing this function and providing solutions. The overall methodological approach is well-structured.

3. The paper covers a comprehensive range of experimental perspectives, including performance in two dimensions of generation, the impact of increasing diversity on performance, and performance variations under different task settings, conducting thorough experiments.

**Weaknesses:**

1. The paper explains in the Introduction how existing reinforcement learning methods improve the diversity of generation at the lexical level but lacks an analysis comparing them with methods like instruction tuning. Instruction tuning uses the same training data template to enable LLMs to learn different tasks, naturally enhancing the model's ability to generate diverse downstream tasks or stylistic language. Since this paper primarily discusses the issue of generation diversity, although it focuses mainly on reinforcement learning, I believe it is still necessary to explore the necessity or fundamental differences compared to methods like instruction tuning.

2. The paper proposes a training method based on determinantal point processes (DPPs) to enhance the generation quality and diversity of LLMs but lacks motivation for adopting DPPs. DPPs have been used in recommendation systems to simultaneously ensure the relevance and diversity of recommended results. Although the authors provide a definition of DPPs in Section 2.3, I still believe the authors need to more clearly explain the principle of how DPPs can improve both aspects simultaneously and the motivation for using them to train LLMs in the Introduction.

3. Some of the figures in the paper lack necessary data labels, dimension identifiers, and summary titles (e.g., Figure 3), requiring readers to read extensive text to understand the content of the figures. Some tables (e.g., Table 1 and Table 2) also lack necessary highlights to allow readers to quickly draw performance conclusions.

4. The paper draws conclusions by comparing the DQO method with baseline methods on general language model tasks, showing that DQO can simultaneously ensure performance in both quality and diversity. However, it lacks a detailed analysis of the differences between the methods. For instance, why do GRPO-likelihood (on the GSM8K task) and GRPO-entropy (on the Dolly task) still perform better in diversity than DQO? Since DPO incorporates considerations for diversity compared to the baseline, why is the improvement in diversity performance not significant for DPO, while there is an improvement in generation quality?

**Questions:**

See questions in weaknesses.

---

> ### Author Response · Authors · 2025-11-23
>
> We appreciate the reviewer’s efforts and positive feedback on our work. We provide the detailed elaboration to your comments below, hope it will solve your concerns and possibly increase the score.
>
> > Comparison with instruction-tuning methods:
>
> Thanks for this point. We are a little bit confused at the comment on "Instruction tuning uses the same training data template to enable LLMs to learn different tasks, naturally enhancing the model's ability to generate diverse downstream tasks or stylistic language". Could you elaborate more on what exact methods you are referring to?
>
> From our understanding, instruction tuning ensures that the model follows instructions (possibly with different styles). We agree with the reviewer that the instruction model induces diversity, however the specific performance in specialized tasks is not guaranteed and the diversity mainly comes from different styles.
>
> Additionally the two approaches are fundamentally different: since instruction tuning requires the access to an input and corresponding output where ours is RL with explicit regularization. However, we will update the paper to provide a comparison discussion for better exposition.
>
> > Motivation of adopting dpps for diversity:
>
> The motivation for using DPPs particularly for promoting diversity, stems from their elegant mathematical formulation which naturally models repulsion between items. We introduced its concept in section 2.3 as well as an intuitive example to illustrate why DPPs are well-suited for enhancing diversity. DPPs have been used in recommendation systems because it is a good model for diversity at first and easy to incorporate other measures such as quality.
>
> > Figure presentation issues:
>
> Thanks for pointing these issues out. We are sorry we didn’t display the table and figure in a reader friendly way. We will definitely increase the font size, add more explanations and highlight the key takeaways in the revision.
>
> > Explanation on behaviors of baselines:
>
> Both GRPO-entropy and GRPO-likelihood promote diversity (through the objective to increase entropy of responses, reward low-likelihood responses respectively), which explains why their diversity metrics are comparable to—or sometimes slightly higher than—those of GRPO. However, in both cases reported by the reviewer, DQO results in significantly higher quality metrics at slightly lower diversity, managing to balance quality and diversity well. It does so consistently across tasks, while other methods fluctuate a lot more.

---

### Official Review · Reviewer_BeWa · 2025-11-01

**Soundness:** 3
**Presentation:** 3
**Contribution:** 3
**Rating:** 6
**Confidence:** 3

**Summary:**

This paper addresses output diversity degradation in large language models (LLMs) following post-training with reinforcement learning methods like PPO and GRPO. The authors propose Diversity Quality Optimization (DQO), a novel training framework that jointly optimizes for response quality and semantic diversity. DQO introduces a Determinantal Point Process (DPP)-based diversity objective in the RL training loss. For each prompt, the method samples a set of responses, maps them into a semantic embedding space using a pretrained encoder, and computes a kernel-based similarity matrix. The diversity score corresponds to the determinant of this matrix. To address computational challenges, the authors introduce two key modifications: (i) adding an identity matrix for regularization, and (ii) employing leave-one-out (LOO) gradient estimators to reduce variance. The authors evaluate DQO across diverse tasks, including instruction-following, summarization, story writing and reasoning.

**Strengths:**

- The paper addresses an important and timely problem in LLM post-training, where training with reinforcement learning often reduces the diversity of generated output.
- The proposed method can be easily integrated with existing RL algorithms, showing potential for broad applicability in post-training methods.
- The experimental evaluation covers multiple task categories and shows consistent improvements in diversity metrics while maintaining comparable quality.

**Weaknesses:**

While the paper is clearly written and the method is well-justified, it could be further strengthened through additional experimental analyses.

- As acknowledged by the authors, the proposed approach relies heavily on the quality of the pretrained embedding model used to compute semantic similarity. Evaluating DQO with different embedding encoders would strengthen the empirical analysis and clarify its generalizability. How does performance vary with different embedding models (e.g., larger models, task-specific embeddings)? Are certain embedding models more suitable for particular task types?
- While the LOO estimator is introduced to mitigate variance, the paper lacks ablation study to isolate its effect on performance or stability.
- Experiments focus only 2 models (Qwen2.5-MATH-1.5B for reasoning, Llama3.2-1B for other tasks). No exploration of different model families. It's unclear whether findings generalize to larger models. How does model capacity affect the quality-diversity trade-off?
- The paper fixes the reward model (Skywork-Reward-V2) across all tasks. How sensitive is DQO to reward model quality or choice?

**Questions:**

- How is the quality score evaluated for generation tasks ssuch as Dolly, summarization and story-writing?
- What is the k value in the main experiment (Table 1)?

---

> ### Author Response · Authors · 2025-11-23
>
> We appreciate the reviewer’s positive feedback and valuable comments on our work. We provide the detailed explanations and additional results below. Hope our elaboration solves your concerns and we kindly request you to consider increasing the score.
>
> > Evaluation across different embedding models:
>
> Thanks for this point. We did ablation studies on four additional embedding models which are shown below. The results show the performance of DQO is robust across different embeddings and tasks. We didn't observe clear evidences that certain embedding models are more suitable for particular task types. We will update the paper to include the ablation results of different embedding models. As for the task-specific embedding, there are only embeddings for a specific task category. We believe it will be more interesting to joint-train the embedding and the policy. In this way, one can possibly train a task-specific embedding that may further improve the results which we left as an interesting future work.
>
> | Embedding / Task | distinct-1 | distinct-4 | self-bleu | self-rouge | pass@1 | pass@10 |
> | :--- | :---: | :---: | :---: | :---: | :---: | :---: |
> | **Dolly** | | | | | | |
> | all-MiniLM-L6-v2 (default) | 0.28 | 0.69 | 0.46 | 0.54 | 5.92 | 8.74 |
> | all-mpnet-base-v2 | 0.29 | 0.72 | 0.47 | 0.55 | 5.75 | 8.57 |
> | gte-small | 0.30 | 0.68 | 0.47 | 0.53 | 5.89 | 8.50 |
> | **GSM8K** | | | | | | |
> | all-MiniLM-L6-v2 | 0.10 | 0.42 | 0.14 | 0.31 | 76.3 | 91.2 |
> | all-mpnet-base-v2 | 0.10 | 0.39 | 0.13 | 0.28 | 76.5 | 90.9 |
> | bge-base-en-v1.5 | 0.09 | 0.39 | 0.14 | 0.28 | 77.4 | 90.4 |
> | **CNN** | | | | | | |
> | all-MiniLM-L6-v2 | 0.25 | 0.75 | 0.41 | 0.61 | 11.08 | 13.02 |
> | all-mpnet-base-v2 | 0.26 | 0.77 | 0.43 | 0.62 | 10.39 | 12.70 |
> | gte-small | 0.25 | 0.74 | 0.41 | 0.60 | 10.55 | 12.89 |
> | **Gen** | | | | | | |
> | all-MiniLM-L6-v2 | 0.45 | 0.96 | 0.77 | 0.74 | 10.44 | 12.36 |
> | gte-small | 0.45 | 0.96 | 0.78 | 0.73 | 10.48 | 12.31 |
> | e5-base-v2 | 0.43 | 0.97 | 0.80 | 0.76 | 10.32 | 12.26 |
>
> > Effects of leave-one-out gradient estimator:
>
> We use leave-one-out (loo) gradient estimator to control the scale of the diversity term. Without loo, it scales up in the order of O(k), while with loo, it scales up with O(log(k)), which stabilizes the training even if k is large. Besides, the loo estimator is proved unbiased, which justifies our design. Loo also enables the use of GRPO algorithm, since without loo, the diversity score will be cancelled out when calculating the empirical advantages in a group manner. Finally, the drawback of using loo is usually the additional cost to sample more data, however, we already sample k responses for the calculation of determinant, hence there is no additional cost to use loo. Overall, we think using loo is a practically and theoretically justified design and it is not a hyperparameter, hence we don’t think there is a need to do ablations on it.
>
> > Different model family and model size:
>
> Llama and Qwen are two main families of open source large language models. From our best knowledge, it is quite common in the community to verify an idea on these two main families of models. For the model size, we agree it is interesting to check the performance on bigger models. We did additional experiments on the model of Qwen/Qwen2.5-7B on GSM8K task. Due to the limited computing resources, we cannot run full tasks at the current stage. The results are shown below.The performance of DQO is consistent and even better on 7B models.
>
> Reward model: Skywork-Reward-V2-Llama-3.2-1B
>
> |   | **distinct-1** | **distinct-4** | **self-bleu** | **self-rouge** | **pass@1** | **pass@10** |
> |------------------|----------------|----------------|---------------|----------------|------------|-------------|
> | GRPO (Qwen2.5-7B) | 0.07 | 0.30 | 0.08 | 0.20 | 81.3 | 89.0 |
> | DQO (Qwen2.5-7B)  | 0.10 | 0.43 | 0.16 | 0.31 | 85.0 | 96.0 |
>
> > Performance on different reward models:
>
> We did an additional experiment on GSM8K task with reward model: Skywork-Reward-V2-Qwen3-0.6B. Skywork-Reward-V2-Qwen3-0.6B is a slightly weaker reward model than Skywork-Reward-V2-Llama-3.2-1B. The immediate effect is the quality of responses, which is reflected in GRPO training. However, DQO shows the robustness to reward models, based on the fact it still significantly boosts the diversity metrics and pass@10 without hurting pass@1 accuracy much, achieving decent quality-diversity balance.
>
> Reward model: Skywork-Reward-V2-Qwen3-0.6B
>
> |   | **distinct-1** | **distinct-4** | **self-bleu** | **self-rouge** | **pass@1** | **pass@10** |
> |------------------|----------------|----------------|---------------|----------------|------------|-------------|
> | GRPO (Qwen2.5-7B) | 0.11 | 0.54 | 0.22 | 0.39 | 75.9 | 92.1 |
> | DQO (Qwen2.5-7B)  | 0.14 | 0.63 | 0.30 | 0.46 | 74.8 | 95.1 |
>
> > Replies to questions:
>
> For generation tasks, the reported quality score is the reward given by the reward model. In Figure 1, the k value is 4.

---

### Official Review · Reviewer_Fy8p · 2025-11-01

**Soundness:** 2
**Presentation:** 3
**Contribution:** 3
**Rating:** 2
**Confidence:** 3

**Summary:**

In this paper, the authors have proposed proposes DQO (diversity quality optimization), a training method that uses DPPs to encourage llms to generate semantically diverse responses while maintaining quality. They do this by defining diversity as the determinant of a kernel matrix constructed from response embeddings. This diversity measure can be incorporated into standard RL objectives like PPO or GRPO as a simple regularizer term in the loss function.

**Strengths:**

1. The paper is well-written with good intuitive explanations.
2. This paper addresses an important problem in RL-finetuning research, i.e. diversity collapse. If the limitations (see below) were addressed, this could be useful for the community.

**Weaknesses:**

1. I strongly feel that the reward hacking failure with outcome rewards (in appendix E) should be on the main submission. This isn't a minor edge case. Such outcome rewards are standard for mathematical reasoning or coding tasks. Given that the attempted fix using prompt-response similarity also failed, this issue severely limits the method's applicability and should be acknowledged in the main experiments.
2. Claims about "semantic diversity" are never validated with human judges, and even for LLM-as-judge there's only one model used. I feel that the authors should take a subset of generations and atleast show some correlation between human responses and their LLM-as-judge.
3. Figure 4 shows pass@n curves but the differences are often small and lack error bars. Also (minor), please increase the text size for fig 4 and 5.
4. What happens when you don't set β=α? The derivation in Eq. 5 requires this, but it's a strong constraint. This constraint means you can't independently tune:
a. How much you penalize deviation from the reference policy (β)
b. How much you reward diversity (α)
If β controls exploration vs. exploitation in the standard RL sense, and α controls quality vs. diversity tradeoff, being forced to set them equal is quite restrictive. In practice, you might want β=0.01 for conservative updates but α=1.0 for strong diversity. So why was such a constraint added?

**Questions:**

1. What happens with 7B or 13B models? Does the diversity-quality tradeoff change?
2. How expensive is computing k embeddings and determinants per update?
3. Can you run a human study (even small-scale) where annotators judge semantic diversity? This is essential for validating the core claim of this paper.
4. How sensitive is DQO to the underlying embedding model?
5. Does improved diversity during training lead to better out-of-distribution generalization or just different in-distribution behavior? Maybe choose an OOD summary task for this.

---

> ### Author Response · Authors · 2025-11-23
>
> We appreciate the reviewer’s efforts and valuable comments on our work. We are confident that we can address the weaknesses raised by the reviewer. And we kindly request you to consider increasing the score based on our detailed explanation and additional results.
>
> > Reward hacking with outcome rewards:
>
> We agree that outcome based rewards are very common, but so are reward models. Many aspects of quality cannot be captured with outcome-based rewards and reward modeling still forms a crucial part in the training pipeline of state-of-the-art Language models. We mentioned this phenomenon in the main page and due to the page limitation, we leave the detailed discussion to the appendix. But we are happy to move the detailed discussion into the main body of the paper. Note that it is an inherent issue with outcome reward and is outside the scope of this paper, which we view as an interesting and important direction for future work.
>
> > Diversity evaluation with human judges:
>
> We know diversity is hard to evaluate simply from some metrics, that's one of the reasons why we did toy experiments on the city recommendation task. In this task, the diversity of the outputs are straightforward. And the results in Figure 3 can evidently show the effectiveness of DQO.
>
> We understand that human surveys can give a more reliable evaluation, but it requires human resources and is inconvenient.
> However, we still did a small scale human survey on judging the diversity of the responses generated by the DQO model and baseline model. We randomly select 18 prompts from the test set of four benchmarks (five per task, except summarization, for which we select three to keep the evaluation time manageable). Each prompt is followed by 4 responses from each model. The participant can give a dense score from 0 to 1 to show the preference of a model. And the order of models are shuffled, so there is no information leakage. The results are shown below: each column is a prompt, and we show the preference score of the DQO model. From the results we can see both LLM-as-a-judge and human highly rated DQO as the more diverse one. And the human study results are highly correlated with the LLM-as-a-judge evaluation, consistent on most prompts.
>
> |   |   |   |   |   |   |   |   |   |   |   |   |   |   |   |   |   |   |   |
> |---|---|---|---|---|---|---|---|---|---|---|---|---|---|---|---|---|---|---|
> | LLM-as-a-judge | 0.7 | 0.8 | 0.7 | 0.8 | 0.55 | 0.4 | 0.6 | 0.6 | 0.5 | 0.8 | 0.9 | 0.7 | 0.8 | 0.6 | 0.6 | 0.75 | 0.7 | 0.6 |
> | Human evaluation | 0.65 | 0.74 | 0.79 | 0.67 | 0.62 | 0.5 | 0.74 | 0.63 | 0.54 | 0.51 | 0.65 | 0.61 | 0.88 | 0.69 | 0.59 | 0.62 | 0.78 | 0.68 |
>
> We also manually checked a decent amount of output from the models to ensure that using LLM-as-a-judge is reliable. LLM-as-a-judge has become a convenient and standard evaluation method in the community. With the additional human study, we believe LLM-as-a-judge is effective and reliable in judging the diversity of model outputs. We will do a bigger scale human survey in the future to further evaluate the performance.
>
> > Constraint on $\alpha=\beta$ in the analysis:
>
> We want to clarify that it is not a constraint, instead, only for the simplicity of exposition, we assume $\alpha=\beta$. There is no need to set $\alpha=\beta$ no matter in theory or practice. In the case when $\alpha=\beta$, we have,
>
> $    \pi_{div}(y_{1:k}|x)\propto \pi_{ref}(y_{1:k}|x) (det(L_\psi(x,y_{1:k})))^{\alpha/\beta} $
>
> where the new embedding vector is defined $\psi(x,y)=\sqrt{exp\left(\frac{r(x,y)}{\alpha}\right)}\cdot\phi(y)$.
> As you have observed, $\alpha$ controls the balance of reward and diversity which is captured in the new feature vectors. And $\beta$ controls the regularization of the reference policy.
>
> > Small difference in pass@n performance:
>
> Note that the goal of DQO is to boost the diversity while ensuring the quality of a single response, in other words, we are not directly optimizing the pass@n performance. Based on this fact, we think the difference in pass@n performance is decent and our model achieves a very good quality-diversity balance. There is always a tradeoff between quality and diversity. One cannot achieve good diversity without sacrificing the quality. Some existing works show that the base model can have a decent pass@n accuracy on math benchmarks when n is very large, but we cannot say base model is preferred since it has low quality of single response.
>
> We apologize for the small font size in some figures. We will correct this and increase the font size in the revision.

---

> > ### Author Response · Authors · 2025-11-23
> >
> > > Different model size performance:
> >
> > We agree it is interesting to check the scalability of our method. Due to the limited computing resources, we cannot run full experiments on all tasks at the current stage. We did additional experiments on the model of Qwen/Qwen2.5-7B on GSM8K task (the setting is consistent with the report in our paper except the training steps, here we train one epoch). The results are shown below. Based on this preliminary result, DQO shows the consistent and even better performance on 7B models.
> >
> > |   | **distinct-1** | **distinct-4** | **self-bleu** | **self-rouge** | **pass@1** | **pass@10** |
> > |------------------|----------------|----------------|---------------|----------------|------------|-------------|
> > | GRPO (Qwen2.5-7B) | 0.07 | 0.30 | 0.08 | 0.20 | 81.3 | 89.0 |
> > | DQO (Qwen2.5-7B)  | 0.10 | 0.43 | 0.16 | 0.31 | 85.0 | 96.0 |
> >
> > > Additional computational cost of DQO:
> >
> > The additional computational cost mainly comes from two parts. First is computing the feature vector via embedding models. Note that embedding models are non-autoregressive and the size is relatively small. So the cost of mapping responses to embeddings is cheap. Second is the calculation of the determinant. Given k responses per prompt, to calculate the determinant of the kernel matrix, the complexity is roughly O(k^3). In our setting, k is limited (less than 10). Our ablations on the value of k and $\alpha$ further shows that k does not need to be large. With a small value of k, the computation cost of the determinant is negligible (microseconds). Overall, the additional cost is ignorable. Below, we show the concrete results of the average time taken per training step of DQO compared with the baseline.
> >
> > | Task | GRPO (s/step) | DQO (s/step) |
> > |------------------|----------------|----------------|
> > | Reasoning | 36.844 | 36.216 |
> > | Summarization  | 34.826 | 34.463 |
> > | Instruction-following | 40.962 | 41.485 |
> > | Creative-writing | 20.963 | 22.462 |
> >
> > Note that it is reasonable that DQO can be faster than GRPO considering the variance. Since these are online algorithms, the time depends on the length of sampled responses.
> >
> > > Sensitivity to embedding models:
> >
> > We did ablation studies on four additional embedding models. The results show the performance of DQO is robust across different embeddings. Please refer to our response to Reviewer J8Sx for the exact results.
> >
> > > Out-of-distribution performance:
> >
> > We thank the reviewer for this suggestion. We did two ood experiments. We tested the pass@10 performance on openai/summerize_from_feedback benchmark with the model trained on cnn-dailymail dataset. The pass@10 reward is 8.73 (GRPO) versus 8.81 (DQO). We also tested on MATH500 task with the model trained on GSM8K dataset. The pass@10 performance is 81.9% (GRPO) versus 81.1% (DQO).
> >
> > The result shows DQO maintains ood performance parity with the baseline method but does not yield significant improvement which indicates DQO mainly improves in-distribution behaviors. This phenomenon is reasonable. First, there is no direct correlation between diversity and generalization on ood. Besides, theoretically the DQO objective is designed to encourage the coverage of high-quality modes within the support of the training distribution.

---

> ### Comment · Reviewer_Fy8p · 2025-11-25
>
> I thank the authors for their detailed rebuttal and for answering all my questions with appropriate experiments! Since all my queries are answered, I am increasing my score to 6.
>
> By the way, I think the authors made a minor typo here in their rebuttal:
> >  In the case when α=β, we have,
>
> Shouldn't it be "α != β"?

---

> > ### Author Response · Authors · 2025-11-27
> >
> > Yes, it is a typo, it should be $\alpha \neq \beta$. Thanks for pointing it out and your positive feedback on our rebuttal.

---

### Official Review · Reviewer_J8Sx · 2025-11-01

**Soundness:** 4
**Presentation:** 3
**Contribution:** 4
**Rating:** 6
**Confidence:** 4

**Summary:**

This paper introduces a post-training method called DQO (Diversity Quality Optimization) that is designed to jointly optimize LLMs for quality and semantic diversity. Using Determinantal Point Processes, the authors gather the embeddings of multiple responses from each prompt and measure diversity as the volume spanned by these embeddings by taking the determinant of their pairwise similarity matrix. This diversity measure is added to the objective function for RL post-training. DQO maintains a higher quality-diversity Pareto frontier, matches the baseline for pass@1 and scales better than the baseline for pass@n >1.

**Strengths:**

- Well written

- Clear presentation

- Very good motivation and an effective solution to sustaining diversity

- The determinantal point process is a explained clearly and can have a lot of potential in other pipelines that use frozen embeddings

**Weaknesses:**

## Minor

- The intro reads more like related works. Could be shaped to flow better.

- Incredibly small fonts in the axis labels

- Missing citation semantic entropy [1]

## Major

- **Matrix via similarity Kernel Bottleneck**: There is a k by k matrix to find the determinant of. Is it cubic complexity? Is there a way to mitigate that? How much more is the overhead? A computational cost comparison is pertinent.

- **Reward Model reliance**: The paper is introduced as a general post-training solution but it still requires a good frozen embedding model and a good reward model. Can this method even be reasonably applied to the vast space of situations where those are not available?

- **Embedding Model**: It is necessary to see that the results hold for different embeddings models and embedding approaches. For example something like the Matryoshka representations [2] would be really helpful in supporting the generalizability of this method.


[1] Kuhn, Lorenz, Yarin Gal, and Sebastian Farquhar. "Semantic uncertainty: Linguistic invariances for uncertainty estimation in natural language generation." arXiv preprint arXiv:2302.09664 (2023).

[2] Kusupati, Aditya, et al. "Matryoshka representation learning." Advances in Neural Information Processing Systems 35 (2022): 30233-30249.

**Questions:**

Q1. Can you explain how equation 4 and 5 follow from 3?

Q2. Can you get more results with different embedding or reward models? (embeddings models most importantly).

Q3. Can you provide a description and analysis of the computational cost of the DPPs

---

> ### Author Response · Authors · 2025-11-23
>
> We appreciate the reviewer’s positive feedback, and valuable comments on our work. We provide the detailed elaboration to your comments below, hope it will solve your concern and possibly increase the score.
>
> > Ablations/reliance on embedding and reward models:
>
> Thank the reviewer for this suggestion. We did further ablations on different embedding models (the results are shown below, the setting is same as in the paper with $k=4$, $\alpha=1.0$). From the ablations results, it is evident that our proposed method DQO is robust to different embedding models. We will update the paper to include the ablation of embedding models.
>
> | Embedding / Task | distinct-1 | distinct-4 | self-bleu | self-rouge | pass@1 | pass@10 |
> | :--- | :---: | :---: | :---: | :---: | :---: | :---: |
> | **Dolly** | | | | | | |
> | all-MiniLM-L6-v2 (default) | 0.28 | 0.69 | 0.46 | 0.54 | 5.92 | 8.74 |
> | all-mpnet-base-v2 | 0.29 | 0.72 | 0.47 | 0.55 | 5.75 | 8.57 |
> | gte-small | 0.30 | 0.68 | 0.47 | 0.53 | 5.89 | 8.50 |
> | **GSM8K** | | | | | | |
> | all-MiniLM-L6-v2 | 0.10 | 0.42 | 0.14 | 0.31 | 76.3 | 91.2 |
> | all-mpnet-base-v2 | 0.10 | 0.39 | 0.13 | 0.28 | 76.5 | 90.9 |
> | bge-base-en-v1.5 | 0.09 | 0.39 | 0.14 | 0.28 | 77.4 | 90.4 |
> | **CNN** | | | | | | |
> | all-MiniLM-L6-v2 | 0.25 | 0.75 | 0.41 | 0.61 | 11.08 | 13.02 |
> | all-mpnet-base-v2 | 0.26 | 0.77 | 0.43 | 0.62 | 10.39 | 12.70 |
> | gte-small | 0.25 | 0.74 | 0.41 | 0.60 | 10.55 | 12.89 |
> | **Gen** | | | | | | |
> | all-MiniLM-L6-v2 | 0.45 | 0.96 | 0.77 | 0.74 | 10.44 | 12.36 |
> | gte-small | 0.45 | 0.96 | 0.78 | 0.73 | 10.48 | 12.31 |
> | e5-base-v2 | 0.43 | 0.97 | 0.80 | 0.76 | 10.32 | 12.26 |
>
> We also did the ablations on an additional reward model on GSM8K task. Please refer to our response to Reviewer BeWa for the concrete results.
>
> As pointed out, we assume the access to embedding and reward models. However, we don't think it is a major limitation considering the popularity and accessibility of these general ability embedding/reward models nowadays. At the same time, we agree that it would be interesting to get rid of the requirement of an existing embedding model and joint-train the embedding and the policy. In this way, one can possibly train a task-specific embedding that may further improve the results which we left as an interesting future work.
>
> > Computational cost analysis:
>
> The computational cost of the determinant computation is indeed $O(k^3)$ where k is the number of answers generated per prompt. However, since k is generally small (<10), it requires on the order of a few thousand floating-point operations which is cheap on any modern CPU/GPU (microseconds), especially using the standard LU factorization also used in np.linalg.det().
> And our ablations on the value of $\alpha$ (the coefficient balance reward and diversity) and k further supports the claim. The effect of increasing the value of alpha is similar to increasing k, so practically it is enough to choose a small value of k, which makes the calculation of determinant negligible.
>
> The true computational head comes from using the embedding model to map the response to feature vectors. But since embedding models are not autoregressive models and the size is relatively small, it does not cost much either.
> We provide the concrete results of the average time taken per training step of DQO compared with the baseline below.
>
> | Task | GRPO (s/step) | DQO (s/step) |
> |------------------|----------------|----------------|
> | Reasoning | 36.844 | 36.216 |
> | Summarization  | 34.826 | 34.463 |
> | Instruction-following | 40.962 | 41.485 |
> | Creative-writing | 20.963 | 22.462 |
>
> We can see from the time statistics that the computational burden is negligible. The average training time of DQO is almost same as the baseline. Note that it is reasonable that DQO can be faster than GRPO considering the variance. Since these are online algorithms, the time depends on the length of sampled responses.
>
> > Derivation of (4),(5) from (3):
>
> Derivation of (4) from (3) comes from the following simplified fact,
>
> $E_{y\sim \pi}\left[r(y)\right] -\beta KL(\pi_\theta||\pi_{ref})=-\beta KL(\pi(\cdot)||{\pi_{ref}(\cdot)\exp(r(\cdot)/\beta)})$,
>
> this gives us the optimal formula of $\pi$ which is proportional to $\pi_{ref}(\cdot)\exp(r(\cdot)/\beta)$. Regarding the whole term of $\sum_{i=1}^k r(x,y_i) + \alpha \log \det(L_\phi(y_{1:k}))$ as the new reward and following the same derivation, one can get (4).
>
> To derive (5) from (4), we define a new feature vector $\psi(x,y)=\sqrt{exp\left(\frac{r(x,y)}{\alpha}\right)\pi_{ref}(y|x)}\cdot\phi(y)$, with this new feature vector, one can show (5) following the proof in Appendix B. We also provide a result when $\alpha$ is different from $\beta$, you can refer to our response to Reviewer Fy8p.
>
> > Minor issues:
>
> We thank the reviewer for pointing these issues out. We will definitely improve the presentation of introduction and figures, and add the missing related works in the revision.

---

> > ### Comment · Reviewer_J8Sx · 2025-11-26
> >
> > Thank you for your answer, I will increase my score accordingly.

---

### Author Response · Authors · 2025-12-03
**Summary of Rebuttal Progress Prior to Score Reset**

Dear Area Chair,

Since the scores were reverted to the initial state due to the data leakage incident, we would like to highlight the progress made during the rebuttal phase. Prior to score reset, we successfully addressed the reviewers’ concerns, and two reviewers subsequently raised their scores: Reviewer J8Sx increased the score from 6 to 8, and Reviewer Fy8p increased the score from 2 to 6. Both changes can be verified by their comments followed our rebuttal.

In our rebuttal, we did additional ablation studies on embedding models, showing that our algorithm DQO is robust across different embedding models. Besides, we did a human study evaluating the diversity of responses generated by DQO compared with the baseline model. The result shows strong correlations to LLM-as-a-judge metrics which justifies the use of LLM-as-a-judge as a valid alternative of human evaluation. Furthermore, we provide a concrete analysis supported by the experimental statistics, showing that the additional computational cost of DQO is negligible. Finally, we clarify some misunderstandings on the technical parts. Overall, we addressed all the reviewers concerns and questions.

Best,
Authors

---

### Meta-Review · Area_Chair_ZkjL · 2026-01-07

**Summary:**

This paper proposes a novel reinforcement learning algorithm for improving the diversity of LLMs based on determinantal point processes (DPPs).

**Reviewer Concerns:**

The reviewers had a wide range of concerns. The most important ones centered around the comprehensiveness of the experimental analysis; in addition, there were a number of clarifying questions about the methodology. The authors provided a number of experiments to address concerns about generality and applicability, as well as clarifications about their methodology. Their response helped address many of the reviewer comments.

**Reviewer Scores:**

The reviewers commented that they increased their score, with the final scores all being positive about acceptance.

---

### Decision · Program_Chairs · 2026-01-26

Accept (Poster)